# Hybrid feature-selection and diversity-guided stacking framework for interpretable ensemble learning: Application to COVID-19 mortality prediction

**Farideh Mohtasham**[1⊙], **Seyed Saeed Hashemi Nazari**[2⊙],
**Mohamad Amin Pourhoseingholi**[3‡], **Kaveh Kavousi**[4*], **Mohammad Reza Zali**[1‡]

**1** Gastroenterology and Liver Diseases Research Center, Research Institute for Gastroenterology and Liver Diseases, Shahid Beheshti University of Medical Sciences, Tehran, Iran, **2** Department of Epidemiology, School of Public Health & Safety, Shahid Beheshti University of Medical Sciences (SBMU), Tehran, Iran, **3** National Institute for Health and Care Research (NIHR) Nottingham Biomedical Research Center, Hearing Sciences, Mental Health and Clinical Neurosciences, School of Medicine, University of Nottingham, Nottingham, United Kingdom, **4** Laboratory of Complex Biological Systems and Bioinformatics (CBB), Department of Bioinformatics, Institute of Biochemistry and Biophysics (IBB), University of Tehran, Tehran, Iran

⊙ These authors contributed equally to this work.
‡ These authors also contributed equally to this work.
* kkavousi@ut.ac.ir

## Abstract

### Background

Reliable predictive modeling in high-dimensional biomedical data requires a balance between accuracy, interpretability, and computational efficiency. However, existing ensemble methods often overlook model diversity or rely on ad hoc feature-selection approaches, which limit generalizability. This study introduces a hybrid feature-selection and diversity-guided stacking framework designed to improve robustness and scalability across clinical and other data-intensive domains.

### Methods

The proposed framework integrates a hybrid feature-selection pipeline—combining Variance Inflation Factor (VIF), Analysis of Variance (ANOVA), Sequential Backward Elimination (SBE), and Lasso regression—to reduce multicollinearity and overfitting. It also employs a diversity-aware stacking strategy that constructs sub-model sets based on pairwise diversity measures (Disagreement, Yule's Q, and Cohen's Kappa) and non-pairwise metrics (Entropy and Kohavi–Wolpert). Sixteen base classifiers and five meta-learners were trained using repeated 10-fold cross-validation. The framework was evaluated using data from 4,778 hospitalized COVID-19 patients with 116 clinical and laboratory attributes, preprocessed using robust scaling and ROSE-based class balancing.

**Data availability statement:** The dataset analyzed in this study contains sensitive patient information and cannot be shared publicly due to confidentiality restrictions. Access to de-identified data can be requested through the Shahid Beheshti University of Medical Sciences Ethics Committee (IR.SBMU.RIGLD. REC.1401.032) at urm@sbmu.ac.ir, which will review and, if appropriate, authorize data release. The corresponding author can provide further details on the application process. All analysis codes developed in this study are available from the corresponding author upon request.

**Funding:** The author(s) received no specific funding for this work.

**Competing interests:** The authors have declared that no competing interests exist.

## Results

The optimal configuration, which stacked Random Forest and XGBoost models using a Neural Network meta-learner, achieved 91.4% accuracy (95% CI: 89.8–92.8), AUC = 0.955, F1 = 0.801, and MCC = 0.746, outperforming the best individual model (AdaBoost, 90.2%). Training time (~450 s) and per-case inference time (<0.2 s) demonstrated computational feasibility. Feature-importance analysis and SHAP-based interpretation confirmed clinical relevance and interpretability.

## Conclusions

The hybrid feature-selection and diversity-guided stacking framework improves predictive accuracy and interpretability while maintaining computational efficiency. Although validated using COVID-19 mortality data, the approach is broadly applicable to biomedical, environmental, and engineering prediction tasks that require interpretable and scalable ensemble learning.

## 1 Introduction

Machine learning (ML) has become an essential component of modern biomedical research, enabling the discovery of complex, nonlinear patterns and supporting improved risk stratification across diverse clinical domains. The primary goal of ML research is to develop efficient, interpretable, and generalize algorithms capable of delivering reliable performance across heterogeneous datasets [1]. Efficiency in ML encompasses not only time and memory requirements but also data utilization and interpretability—key considerations for deployment in high-stakes clinical environments, where transparency, reproducibility, and auditability are critical.

Traditional ML models, however, often encounter significant challenges, including data quality issues, overfitting, class imbalance, and limited interpretability, which restrict their usefulness in clinical applications. These limitations have driven growing interest in ensemble learning, in which multiple algorithms are combined to reduce both variance and bias, thereby improving predictive performance compared with individual models [2–4]. Among ensemble approaches, stacking has emerged as a particularly powerful technique. Stacking integrates diverse base learners and employs a meta-learner to combine their predictions, leveraging complementary error structures to improve accuracy and robustness [5–7].

Recent advancements in various disciplines highlight both the promise and the challenges of ensemble modeling. In cardiovascular diagnostics, Feng et al. (2023) developed a hybrid model that combined hemodynamic modeling with ML to achieve >90% accuracy in under two seconds per case, demonstrating that hybrid approaches can deliver high computational efficiency while maintaining strong performance [8]. Similarly, Wang et al. (2023) used the Super Learner algorithm to improve prediction of cumulative lead exposure, though at the cost of substantial computational load due to the combination of multiple algorithms [9]. In proteomics and

chemical sciences, feature selection and dimensionality reduction have proven effective in enhancing prediction accuracy while reducing model complexity.

Xu et al. (2022) benchmarked 13 ML models for protein-level inference from RNA features across more than 2,500 samples and 20 datasets. Their findings demonstrated that combining appropriate feature selection with classical models and voting ensembles improved accuracy, although computation time varied widely [10].

Reda et al. (2023) applied variable selection with partial least squares regression to predict olive-oil quality parameters using near-infrared spectroscopy, showing that variable reduction improved accuracy and decreased computational demands [11].

In oncology, stacking and hybrid ensembles have also yielded substantial gains in predictive performance and generalization. Mohammed et al. (2021) applied a CNN-based stacking ensemble to multi-cancer RNA-Seq classification, achieving superior accuracy to single models while retaining computational feasibility [12]. Wang et al. (2025) designed a multimodal stacking framework that integrated radiomics and deep learning for head-and-neck cancer prognosis (C-index = 0.93), demonstrating the influence of meta-learner selection on scalability [13]. Kwon et al. (2019) found that gradient boosting performed best as a meta-learner for accuracy, while generalized linear models minimized error in breast cancer classification, highlighting the trade-offs between model complexity and efficiency [14]. Other architectures, such as the relevance-aware capsule network [15], deep convolutional neural networks [16], and U-Net–based MRI segmentation models [17], have demonstrated that improvements in accuracy commonly require substantially higher training time and memory, emphasizing the need to balance predictive strength with practicality and interpretability.

Ensemble learning has been widely adopted in other clinical areas. Abualnaja et al. [18] analyzed 32 studies involving 142,459 patients with meningiomas and reported that combined radiomic and clinical ensemble models achieved AUCs of 0.74–0.81, demonstrating robust multimodal representation. Likewise, Lei et al. [19] analyzed 32 studies involving 142,459 patients with meningiomas and reported that combined radiomic and clinical ensemble models achieved AUCs of 0.74–0.81, demonstrating robust multimodal representation. Other studies in cardiovascular disease have shown similar results. Dhingra et al. [20] developed an ensemble model (PRESENT-SHD) using 261,228 ECGs, achieving AUROC values of 0.85–0.90 across multiple hospitals, indicating strong cross-population stability. Tseng et al. [21] used XGBoost and random forest models to predict acute kidney injury following cardiac surgery (AUC = 0.843), demonstrating ensemble learning's value in perioperative risk prediction.

In infectious diseases research, Sawesi et al. [22] reviewed 17 leptospirosis studies and found that ML and deep learning methods—including CNN ensembles—achieved high accuracy (80–98%), though most lacked external validation. Chiasakul et al. [23] reported that AI methods for venous thromboembolism prediction outperformed traditional risk scores (mean AUC 0.79 vs 0.61), although many studies exhibited bias and limited generalizability.

Ensemble learning has also consistently outperformed single models in forecasting outbreaks of dengue, influenza, Ebola, and COVID-19. Early COVID-19 mortality forecasts demonstrated that ensembles delivered greater accuracy and precision than individual models [24].

Stacking is particularly suited to heterogenous clinical datasets, such as COVID-19 mortality prediction, which depends on numerous clinical, biochemical, and physiological indicators [25]. Berliana and Bustamam [4] demonstrated that a two-level stacking model achieved more than 97% accuracy with CT data and 99% with chest X-ray images. Cui et al [26] introduced a nested heterogeneous ensemble integrating SVR, ELM, and logistic regression, achieving improved generalization. Li et al. [27] predicted early mortality using five base models and a genetic-algorithm optimization procedure, achieving an AUC of 0.907 in a cohort of 4,711 patients. Other studies demonstrated that hybrid ensembles incorporating supervised and unsupervised learning improved performance by over 10%, and that boosted models remained competitive with strong clinical relevance [28,29].

Despite these advances, several methodological limitations persist. Systematic reviews highlight widespread issues such as small and unrepresentative datasets, weak handling of missing data, lack of external validation, and overreliance

on discrimination metrics alone [29–31]. Many studies also neglect calibration, effect-size estimation, or fairness analyses, reducing clinical interpretability [32,33]. research shows a nonlinear relationship between predictive gain and computational cost: complex models often deliver higher accuracy but at significant increases in resource consumption [34–37]. While innovations such as subgraph learning [35] and simplified coronary models [8] can mitigate these burdens, a careful balance of accuracy, efficiency, and interpretability remains necessary.

Sample size adequacy is another concern. Many COVID-19 models are trained on datasets too small for their complexity, leading to instability and overfitting [38]. Class imbalance is also common in mortality modeling; although oversampling and weighting strategies are widely used, these must be validated to avoid artificial distortions [39,40].

Furthermore, many ensemble studies rely heavily on tree-based models such as random forest, XGBoost, and LightGBM, limiting diversity and restricting the full advantages of ensemble learning [41,42]. Quantitative measures of diversity—such as Yule's Q, Disagreement, Cohen's Kappa, or Double-Fault—remain rarely used in COVID-19 modeling despite consistent evidence that diversity improves generalization [42–44].

To address these limitations, this study introduces a computationally efficient, diversity-guided stacking ensemble framework that integrates heterogeneous base classifiers and interpretable meta-learners to predict COVID-19 mortality. Our approach incorporates:

1. Hybrid feature-selection using variance inflation factor (VIF) analysis, ANOVA, sequential backward elimination (SBE), and Lasso regression to control multicollinearity and enhance interpretability;

2. Controlled ensemble depth to balance predictive gain and computational feasibility; and

3. Lightweight meta-learners capable of capturing nonlinear dependencies among diverse base learners.

We constructed sub-model ensembles using multiple diversity metrics across 16 machine learning algorithms and assessed model performance using discrimination, calibration, and statistical significance tests, including Wilcoxon, McNemar, and DeLong analyses. Model interpretability was enhanced through SHAP-based explanation of global and local prediction behavior.

This study presents a generalizable diversity-aware ensemble framework designed to balance accuracy, interpretability, and computational cost. Although applied here to COVID-19 mortality prediction, the approach is suitable for a wide range of biomedical prediction problems that require robust, interpretable, and scalable machine learning solutions.

## 2 Materials and methods

### 2.1 Overview of the proposed framework

As illustrated in Fig 1, this study adopts a multi-stage framework for predicting mortality risk, integrating standard machine learning techniques with the proposed algorithmic innovations. The workflow contains two primary layers:

Foundational Stage – Data preprocessing, normalization, and training of base models using established machine learning procedures.

Algorithmic Stage – A diversity-guided stacking ensemble that integrates hybrid feature selection, explicit model diversity assessment, and a comparison of multiple meta-learners to optimize predictive performance, interpretability, and computational efficiency.

Data from 4,778 confirmed COVID-19 cases were cleaned through exclusion of incomplete records, iterative multivariate imputation, and normalization. The hybrid feature-selection process removed multicollinearity using Variance Inflation Factor (VIF), followed by Analysis of Variance (ANOVA), Sequential Backward Elimination (SBE), and Lasso regression to select 15 key predictors.

Sixteen machine learning classifiers were trained using stratified and balanced datasets and evaluated via repeated 10-fold cross-validation. To enhance predictive performance, ensemble sets were constructed based on correlation and

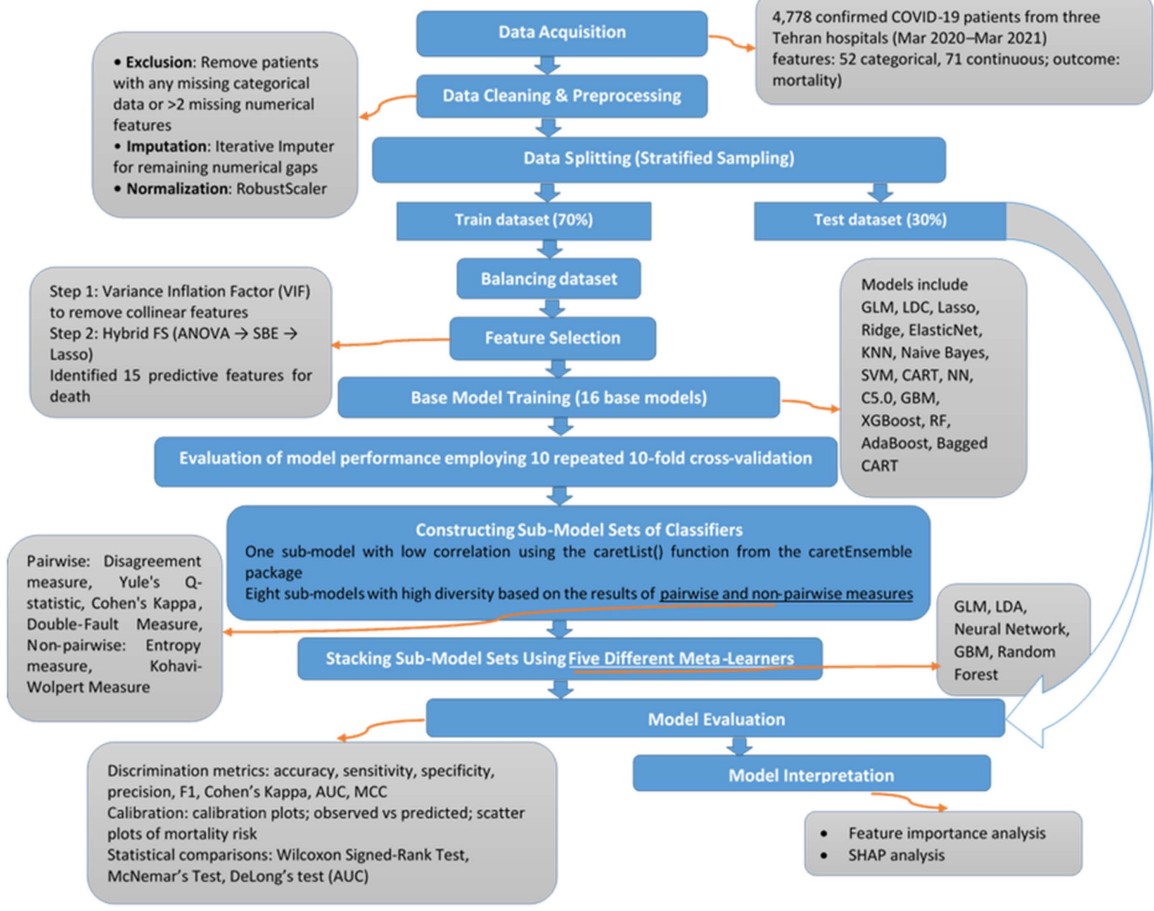

**Fig 1. Research methodology of the proposed machine learning framework.**

statistical diversity metrics, then stacked using five different meta-learners. Models were evaluated based on discrimination and calibration performance and validated using significance tests. Interpretability was assessed using feature importance and SHAP-based analyses.

## 2.2 Data source and ethical approval

Data were obtained from 4,778 confirmed COVID-19 patients admitted to three general hospitals in Tehran, Iran, between March 2020 and March 2021. Demographic, clinical, laboratory, symptom, comorbidity, vital sign, and outcome information was extracted from clinical records reviewed by trained medical staff. Laboratory findings were collected on the first day of admission through the hospital information system, and COVID-19 diagnosis was confirmed using real-time polymerase chain reaction (RT-PCR) of nasal or oropharyngeal swab samples.

The study followed formal institutional requirements and received ethical approval from the Institutional Review Board (IRB) of Shahid Beheshti University of Medical Sciences (IR.SBMU.RIGLD.REC.1401.032).

Informed consent was waived due to the retrospective nature of the study, and data were anonymized prior to analysis in accordance with the Declaration of Helsinki. This dataset provides comprehensive temporal, demographic, and clinical information suitable for developing predictive models for COVID-19 mortality. Additional details on the epidemiological profile of the cohort are available in Hatamabadi et al. [45].

## 2.3  Data preprocessing

Missing data were assessed and addressed prior to model development. Patients with missing values in any categorical variable or more than two missing continuous variables were excluded. Among the 123 available variables (52 categorical and 71 numeric), no categorical variables contained missing values. Seven numerical variables with more than 5% missingness were removed to reduce risk of bias. Remaining missing values were imputed using an iterative multivariate approach implemented in Scikit-learn [46], which models each variable with missing entries as a function of all other features in a chained regression process. This preserves multivariate relationships and minimizes bias under the Missing at Random (MAR) assumption.

The resulting imputed dataset was previously validated by Hatamabadi et al. [47] to confirm realistic variable distributions and consistent multivariate relationships. To account for skewed distributions and sensitivity to outliers, continuous variables were standardized using robust scaling [48], which centers variables on the median and scales using the interquartile range (IQR). This approach enhances stability in clinical models by reducing the influence of extreme values.

## 2.4  Feature selection

Feature Selection (FS) is essential for managing high-dimensional clinical datasets by reducing redundant, irrelevant, or correlated predictors while improving model accuracy, computational efficiency, and interpretability [49,50].

A multi-stage hybrid feature-selection strategy was implemented to progressively eliminate multicollinearity, non-informative predictors, and weak contributors. Four complementary methods were applied:

1. Variance Inflation Factor (VIF): Used to detect and remove highly collinear continuous variables, thereby improving model stability and avoiding inflated variance estimates [51,52].

2. Analysis of Variance (ANOVA): Applied next to evaluate between-group differences using F-tests and eliminate non-discriminative features with minimal computational cost [53,54].

3. Sequential Backward Elimination (SBE): Iteratively removed the least informative features based on cross-validated model performance, preserving meaningful interactions and improving generalization [55].

4. Lasso regression: Imposed regularization to shrink weak coefficients to zero, providing sparse and stable model structures suited for correlated predictors [56].

This integrated pipeline—removing collinearity (VIF), filtering weak predictors (ANOVA), refining via model performance (SBE), and enforcing sparsity (Lasso)—produced a compact, interpretable feature set optimized for ensemble learning.

## 2.5  Model training and selection

Sixteen base machine learning models were trained, including ten standard algorithms and six boosting or bagging methods, representing diverse methodological families. Hyperparameters were optimized through grid search using the caret package [57,58], with 10-fold cross-validation repeated 10 times [59] to balance bias and variance Models with minimal benefit from extensive tuning (e.g., GLM, LDA, CART, Naïve Bayes) retained default settings to maximize computational efficiency.

Hyperparameter ranges were informed by existing literature and empirical results from clinical prediction research. Table 1 summarizes the optimized settings for each model.

Model selection was based on three criteria:

1. Algorithmic diversity,

2. Demonstrated success in biomedical or COVID-19 prediction studies, and

3. Complementary bias–variance profiles.

**Table 1. Parameter settings for the 16 base machine learning models.**

| Models | Tuned hyper parameters |
|---|---|
| **Generalized Linear Model (GLM)** | none |
| **Linear Discriminant Analyses (LDA)** | none |
| **Regularized Regression (Lasso)** | Alpha = 1; lambda = 0.0014 |
| **Ridge Regularized Regression (Ridge)** | Alpha = 0; lambda = 0.098 |
| **Elastic Net Regularized Regression (Elastic Net)** | Alpha = 0.5; lambda = 0.047. |
| **k-Nearest Neighbors (KNN)** | k = 1 |
| **Naïve Bayes (NB)** | none |
| **Support Vector Machine (SVM)** | sigma = 0.15, C = 10 |
| **Classification and Regression Trees (CART)** | none |
| **Neural Network (NN)** | size = 10, decay = 0.1 |
| **C5.0** | trials = 50, model = "tree," winnow = FALSE |
| **Stochastic Gradient Boosting (GBM)** | n.trees = 250, interaction.depth = 5, shrinkage = 0.1, n.minobsinnode = 10 |
| **Extreme Gradient Boosting (XGBoost)** | nrounds = 250, max_depth = 5, eta = 0.4, gamma = 0, colsample_bytree = 0.8, min_child_weight = 1, subsample = 1 |
| **Random Forest (RF)** | mtry = 3, ntree = 700 |
| **AdaBoost Random Forest** | nIter = 100, method = "Adaboost.M1" |
| **Bagged CART (Treebag)** | none |

The final pool (supported by systematic evidence, e.g., Bottino et al. [60]) included linear models (GLM, Lasso, Ridge, Elastic Net), probabilistic models (Naïve Bayes, LDA), instance-based learners (KNN), tree-based models (CART, C5.0, Random Forest, XGBoost, GBM, Treebag), and Neural Networks.

This diversity ensured coverage of linear and nonlinear relationships, uncertainty modeling, and hierarchical interactions common in clinical decision data.

Cross-validated performance guided final model selection, which was subsequently confirmed through independent test-set validation.

## 2.6 Diversity-guided sub-model construction

Traditional stacking often selects base models with low prediction correlation to ensure each contributes complementary information. Highly correlated models (>0.75) add redundancy and weaken ensemble gains [61].

Sixteen candidate models were initially generated using the *caretList()* from the *caretEnsemble* package [57,58]. Prediction correlations were calculated. Where pairs exceeded 0.75 correlation, the less accurate model was removed across 10 repeated 10-fold cross-validation.

To enhance diversity beyond correlation filtering, additional sub-model sets were constructed using explicit diversity metrics capturing complementary error patterns among classifiers. Pairwise measures (Disagreement, Yule's Q, Cohen's Kappa, Double-Fault) and non-pairwise metrics (Entropy, Kohavi–Wolpert) were computed following Tattar [62].

The contingency table defining model agreements ($n_{11}$ and $n_{00}$) and disagreements ($n_{10}$ and $n_{01}$) across $N$ observations is presented in Table 2.

**2.6.1 Disagreement measure.** This quantifies the proportion of instances where the two classifiers differ in prediction(1):

$$DM = \frac{n_{10} + n_{01}}{N}$$

(1)

**Table 2. Contingency table illustrating agreement and disagreement between two classifiers.**

|  | $M_1$ predicts 1 | $M_1$ predicts 0 |
|---|---|---|
| $M_2$ predicts 1 | $n_{11}$ | $n_{10}$ |
| $M_2$ predicts 0 | $n_{01}$ | $n_{00}$ |

Higher values indicate greater diversity and reduced redundancy.

**2.6.2 Yule's Q-statistic.** Yule's Q (or Q-statistic) assesses the strength and direction of association between two classifiers' predictions (range: –1 to +1). Lower absolute values denote weaker association and thus higher diversity(2):

$$\varrho = \frac{n_{11}n_{00} - n_{10}n_{01}}{n_{11}n_{00} + n_{10}n_{01}}$$

(2)

**2.6.3 Cohen's Kappa statistic.** A widely used measure that evaluates inter-model agreement while adjusting for chance. Low or negative Kappa values suggest that classifiers make independent errors, which enhances ensemble robustness.

**2.6.4 Double-fault measure.** Measures the proportion of cases where both classifiers misclassify the same instance. Smaller values indicate complementary error patterns and reduced correlated failures (3):

$$DF = \frac{\sum_{i=1}^{N} I(\widehat{Y}_{i1} \neq Y_i, \widehat{Y}_{i2} \neq Y_i)}{N}$$

(3)

Two non-pairwise metrics were also used:

- **Entropy measure** [63]: Reflects the overall variability of predictions across all classifiers, ranging from 0 (perfect agreement, no diversity) to 1 (maximum diversity).

- **Kohavi-Wolpert Measure** [62]: Derived from error variance decomposition, it quantifies the dispersion of predictions across classifiers; higher values imply greater diversity and richer ensemble representation.

Pairwise metrics identified redundant learners, while non-pairwise metrics assessed overall heterogeneity within sub-model sets. This integrated diversity evaluation ensured complementary base learners and improved the generalization performance of the stacking ensemble.

## 2.7 Meta-learner integration

Predictions from the sub-model sets were integrated using a stacking framework with five meta-learners:

- Generalized Linear Model (GLM)

- Linear Discriminant Analysis (LDA)

- Random Forest (RF)

- Gradient Boosting Machine (GBM)

- Neural Network (NN)

Linear models (GLM, LDA) were chosen for their transparency and stable inference, while tree-based (RF, GBM) and neural meta-learners modeled nonlinear dependencies among base models. This balanced design supports both

interpretability and computational efficiency, aligning with the study's objective of developing a robust, clinically applicable framework for mixed-type healthcare data.

Stacking was implemented using the *caretEnsemble* to effectively fuse diverse predictive outputs.

## 2.8 Model evaluation and statistical analysis

Performance was assessed using an independent test dataset. Discrimination metrics included accuracy, sensitivity, specificity, precision, F1-score, Cohen's Kappa, area under the ROC curve (AUC), and the Matthews correlation coefficient (MCC). These metrics collectively capture overall correctness, class-specific detection, and robustness under class imbalance—critical in mortality prediction, where false negatives carry severe clinical risk and false positives may strain resources. AUC quantified global discrimination across thresholds, while MCC provided a balanced evaluation under uneven outcome distributions [64].

Calibration was assessed using reliability curves to compare predicted probabilities against observed outcomes. Statistical comparisons employed:

- Wilcoxon signed-rank test was applied for accuracy differences,

- Effect size ($r = \frac{z}{\sqrt{N}}$) [65] interpreted per Cohen's criteria (large = 0.5, medium = 0.3, small = 0.1) [66],

- Holm correction to control the family-wise error rate [67],

- McNemar's test to compare classification outcomes between paired models,

- DeLong's test to assess the statistical significance of AUC differences under the null hypothesis of equal performance [68].

All tests were conducted using appropriate R packages, including rstatix [69]and pROC.

## 2.9 Model interpretability

Feature importance and effect analyses were conducted to explain individual predictions and quantify how specific feature values influenced model outputs, using tools from the *iml* package [70]. To further enhance interpretability, we employed SHAP (SHapley Additive exPlanations), a model-agnostic framework that quantifies each feature's contribution to predicted outcomes [71].

SHAP analysis was applied directly to the final stacked ensemble, treating it as a single predictive function. This approach decomposed predicted probabilities into feature-level attributions of the original clinical variables rather than intermediate model outputs, enabling transparent interpretation of global importance, feature interactions, and local instance-level effects driving mortality predictions.

## 3 Results

### 3.1 Data characteristics and preparation

The study analyzed 4,778 confirmed COVID-19 cases, comprising 116 clinical, laboratory, and demographic features. The overall mortality rate was 22% (1,050 patients). Males accounted for 59.6% of deaths. The mean age of deceased patients was 70.8 years (SD = 15.6), compared with 58.3 years (SD = 16.9) among survivors. Mortality showed significant associations with comorbidities such as hypertension, diabetes, and heart failure, highlighting their importance in COVID-19 risk assessment (Fig. 2).

Numeric features were standardized using the robust scaler based on the interquartile range (IQR) to reduce the influence of outliers. The dataset was split into training (70%, n = 3,345) and testing (30%, n = 1,433) subsets, maintaining consistent mortality rates (21.97%) across both. The original training set exhibited class imbalance (22% "Death" vs.

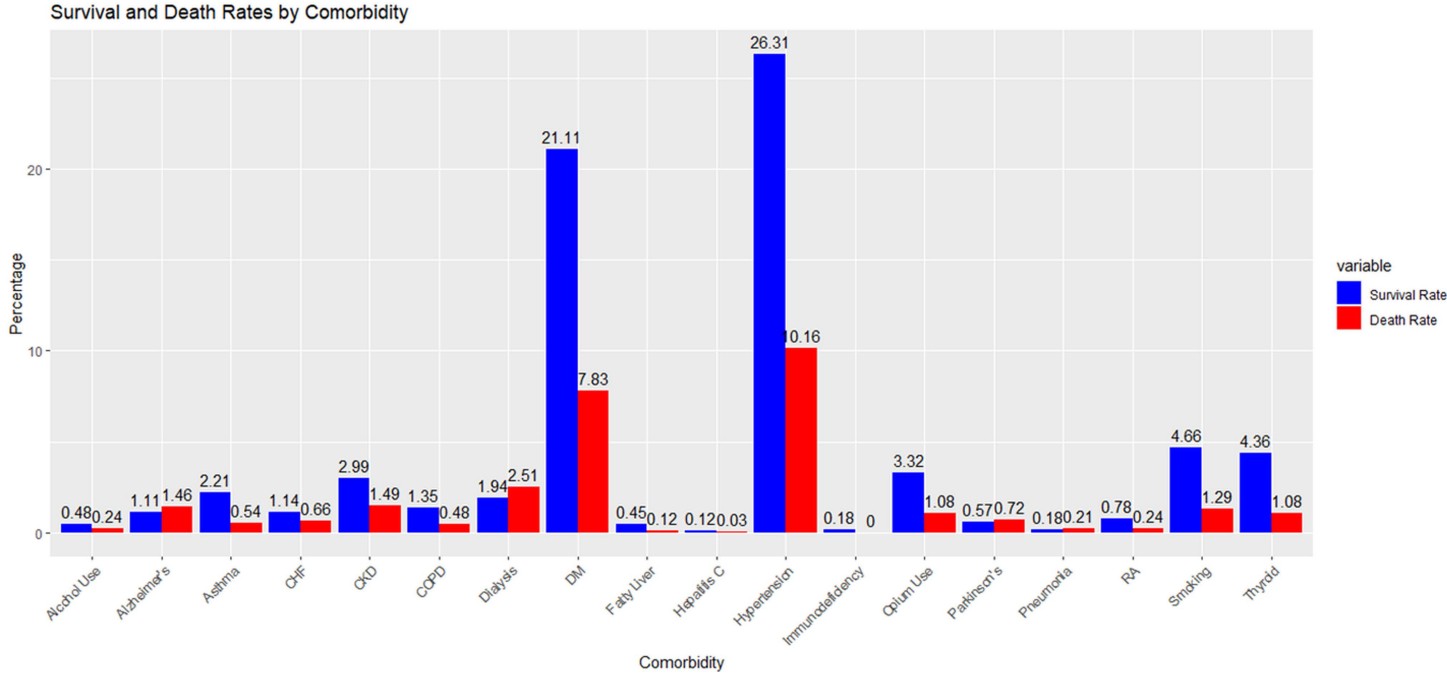

**Fig 2. Comorbidity distribution and mortality associations in the study cohort.**

78% "Alive"), which was corrected using the ROSE package [72], resulting in a balanced 1:1 ratio ("Death": 1,616; "Alive": 1,729). This ensured robust model training and reliable performance evaluation.

### 3.2 Feature Selection and correlation analysis

VIF analysis removed highly collinear variables, reducing the dataset to 109 predictors. ANOVA eliminated 39 features with limited predictive value. SBE and Lasso further refined the selection to 15 key mortality predictors: age, neutrophil count (NEUT), lactate dehydrogenase (LDH), ferritin, phosphorus (P), ventilator oxygen saturation ($O_2$sat.Ventilator), total iron-binding capacity (TIBC), fasting blood sugar (FBS), procalcitonin, serum sodium (Na), muscle pain, chronic kidney disease (CKD), taste/smell loss, D-dimer, and erythrocyte sedimentation rate (ESR).

This selection achieved an optimal balance between model complexity and predictive performance, as adding more features did not improve cross-validation accuracy. The final feature set captured multiple pathophysiological domains relevant to COVID-19 outcomes, including inflammation (NEUT, ESR, ferritin), oxygenation ($O_2$sat.Ventilator), metabolism (FBS, Na, P, TIBC), and organ dysfunction (procalcitonin, CKD). Symptom-based predictors such as muscle pain and loss of taste/smell further enhanced discrimination between severe and non-severe disease.

Correlation analysis (S1–S2 Tables in S1 File) revealed generally weak associations among predictors, indicating low multicollinearity. Moderate correlations were observed between ferritin and ESR ($r = 0.26$), ferritin and TIBC ($r = -0.34$), and TIBC and ESR ($r = -0.35$), reflecting physiologically coherent inflammation–iron metabolism dynamics. Overall low inter-feature correlations support model stability and generalizability.

All selected predictors are routinely collected in clinical settings, ensuring clinical interpretability and feasibility for integration into real-world decision-support systems (**Fig 3**).

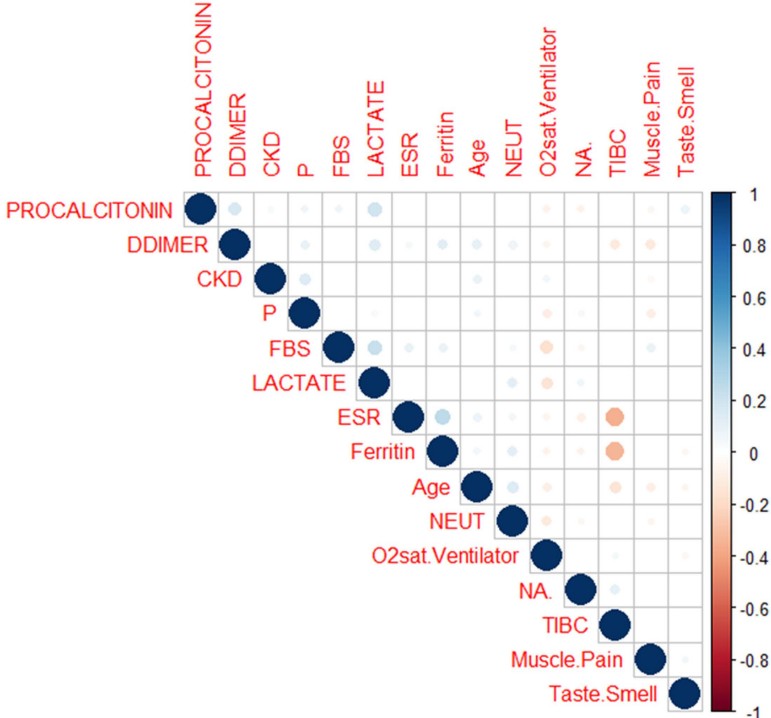

**Fig 3. Correlation between selected features and the outcome (Death/Alive) in the training dataset.**

### 3.3 Model performance evaluation

Fig 4 and Table S3 in S1 File summarize model performance using 10 repeated 10-fold cross-validation across ten base classifiers and six boosting/bagging algorithms applied to all 15 predictive features. Accuracy estimates with 95% confidence intervals indicated that AdaBoost achieved the highest performance, with a mean accuracy of 92.81% (SD = 0.013). Optimized hyperparameters for each models are presented in S1 Fig in S1 File.

### 3.4 Diversity-guided sub-model construction

The first sub-model set was generated using a traditional stacking approach, producing 16 candidate models with the caretList() function. Correlation analysis from repeated 10-fold cross-validation revealed strong dependencies among several learners. The Generalized Linear Model (GLM) demonstrated high correlations with LDA (0.940), Lasso (0.984), Ridge (0.966), and Elastic Net (0.966); similarly, C5.0 and Random Forest (RF) were highly correlated (0.806). AdaBoost also showed high correlations with NN (0.798), GBM (0.848), and XGBoost (0.950). To reduce redundancy, the less accurate model from each correlated pair was removed, retaining four classifiers—Ridge, KNN, CART, and AdaBoost—for the first sub-model set.

To improve model complementarity, additional sub-model sets were constructed using diversity metrics.

Pairwise analyses indicated that GLM, LDA, and other linear models exhibited high agreement, whereas GBM, NN, and RF displayed greater disagreement, suggesting complementary error patterns (Fig 5).

Non-pairwise metrics further quantified ensemble heterogeneity. Entropy values ranged from 0.64 to 0.90, with the NN–GBM combination showing the greatest diversity. Yule's Q ranged from 0.39 to 0.47, while Kohavi–Wolpert values ranged

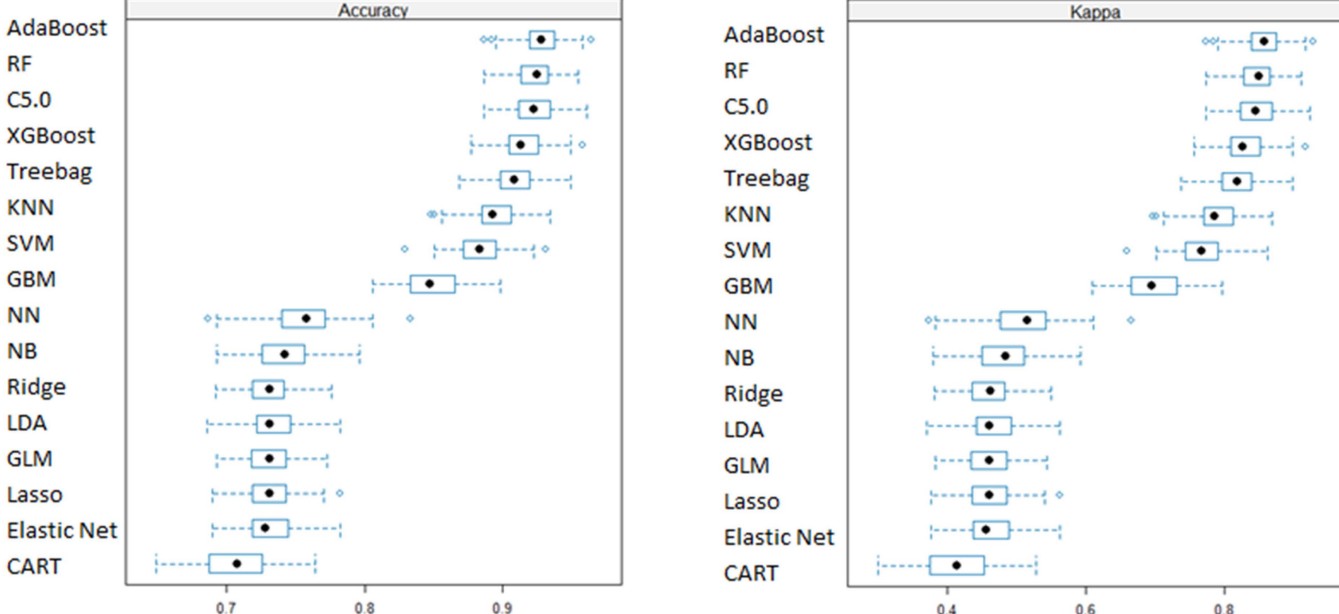

**Fig 4. Performance comparison of base, boosting, and bagging machine learning algorithms using repeated 10-fold cross-validation on the training data.**

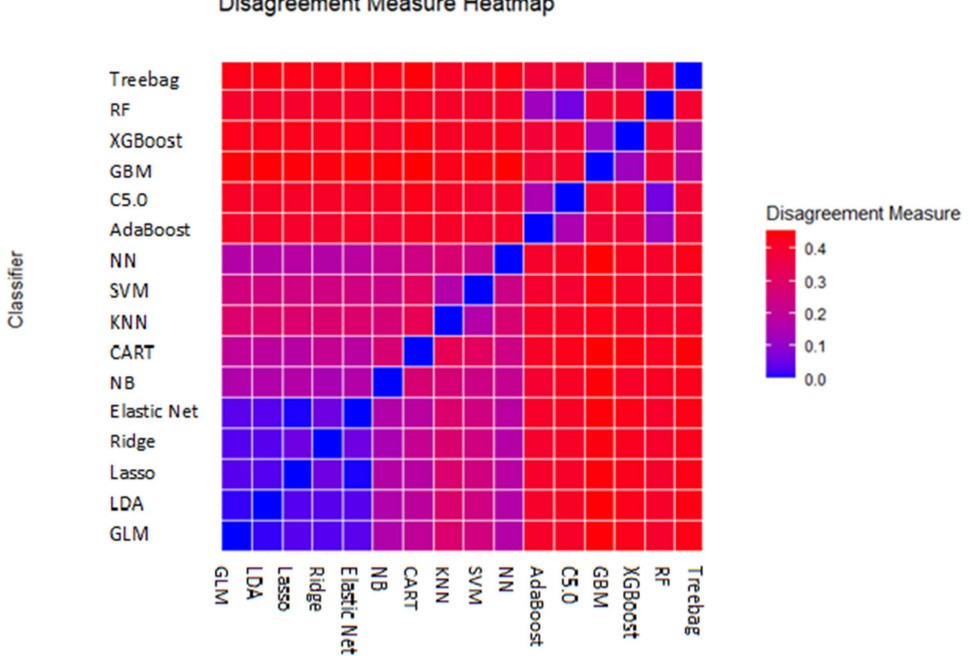

**Fig 5. Disagreement metrics among classifier predictions on the test dataset.**

from 0.099 to 0.195. Negative Kappa values in some sets indicated substantial prediction disagreement, reinforcing diversity among classifiers (Fig 6, S4 Table in S1 File).

Ultimately, eight sub-model sets were selected based on these metrics (Table 3, Fig 7).

This diversity-driven selection ensured inclusion of models differing in both architecture and error behavior, enhancing ensemble robustness and reducing correlated prediction errors.

### 3.5 Stacking model evaluation and statistical comparison

Stacking was performed using five meta-learners: Generalized Linear Model (GLM), Linear Discriminant Analysis (LDA), Neural Network (NN), Gradient Boosting Machine (GBM), and Random Forest (RF). Table 4 summarizes accuracies across stacking configurations on the independent test dataset.

Not all stacking configurations improved upon the strongest base classifier. Ensembles composed of highly correlated models (e.g., Ridge–KNN–CART–AdaBoost) yielded limited gains, indicating performance saturation. In contrast, heterogeneous combinations—such as NB+GBM, RF+XGB, and NB+C5.0+GBM—achieved significant improvements (accuracy up to 0.914). The NN meta-learner consistently outperformed other meta-learners by capturing nonlinear relationships among base-model outputs.

Statistical analyses (Table 5) showed that AdaBoost remained superior to some stacking configurations, supported by significant Wilcoxon results favoring the single model.

Wilcoxon signed-rank, McNemar's, and ROC-based tests compared stacking models with their best-performing base learners, applying Holm-adjusted p-values to control family-wise error and reporting effect sizes (r). Most

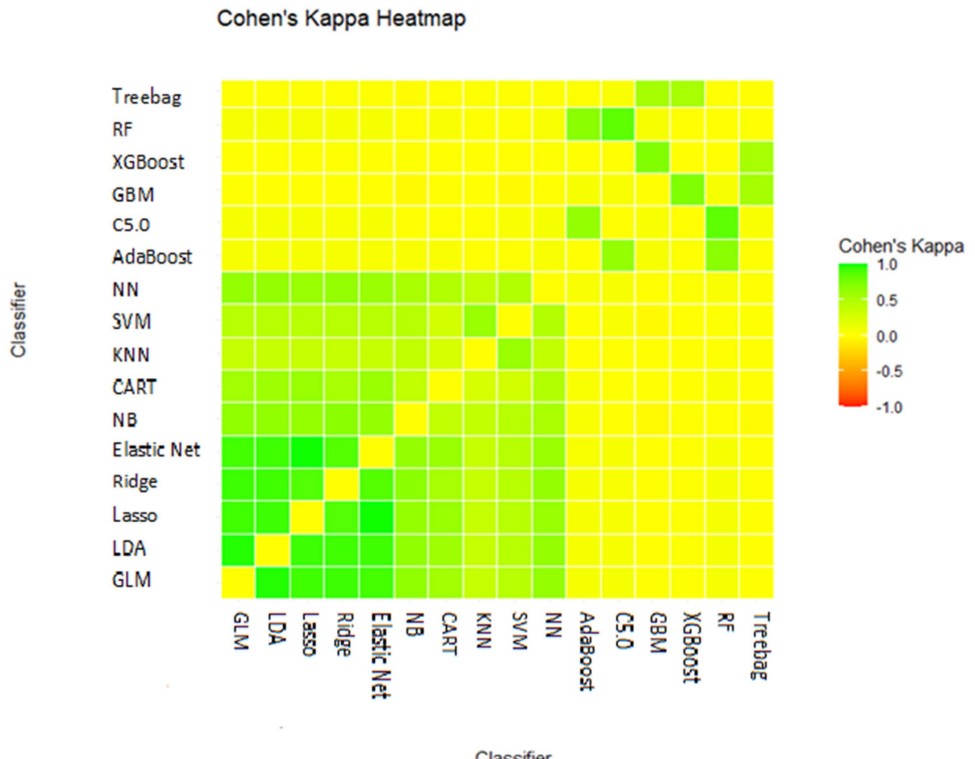

**Fig 6. Inter-rater agreement among classifier predictions on the test dataset.**

**Table 3. Diversity metrics for the eight selected sub-model sets on the test dataset.**

| Sub-Model Sets | Entropy measure | Disagreement Measure (Q statistic) | Kohavi-Wolpert measure | Interrater agreement measure (Kappa) |
|---|---|---|---|---|
| NN, GBM | 0.903 | 0.451 | 0.113 | −0.028 |
| NB, GBM | 0.892 | 0.446 | 0.111 | −0.032 |
| SVM, GBM | 0.881 | 0.440 | 0.110 | −0.082 |
| KNN, AdaBoost | 0.822 | 0.411 | 0.103 | −0.043 |
| RF, XGB | 0.799 | 0.399 | 0.099 | 0.140 |
| NB, CART, AdaBoost, GBM | 0.640 | 0.395 | 0.148 | 0.100 |
| C5.0,NB, GBM | 0.634 | 0.422 | 0.141 | 0.049 |
| NN, RF, CART, GBM, XGB, Treebag | 0.722 | 0.469 | 0.195 | −0.029 |

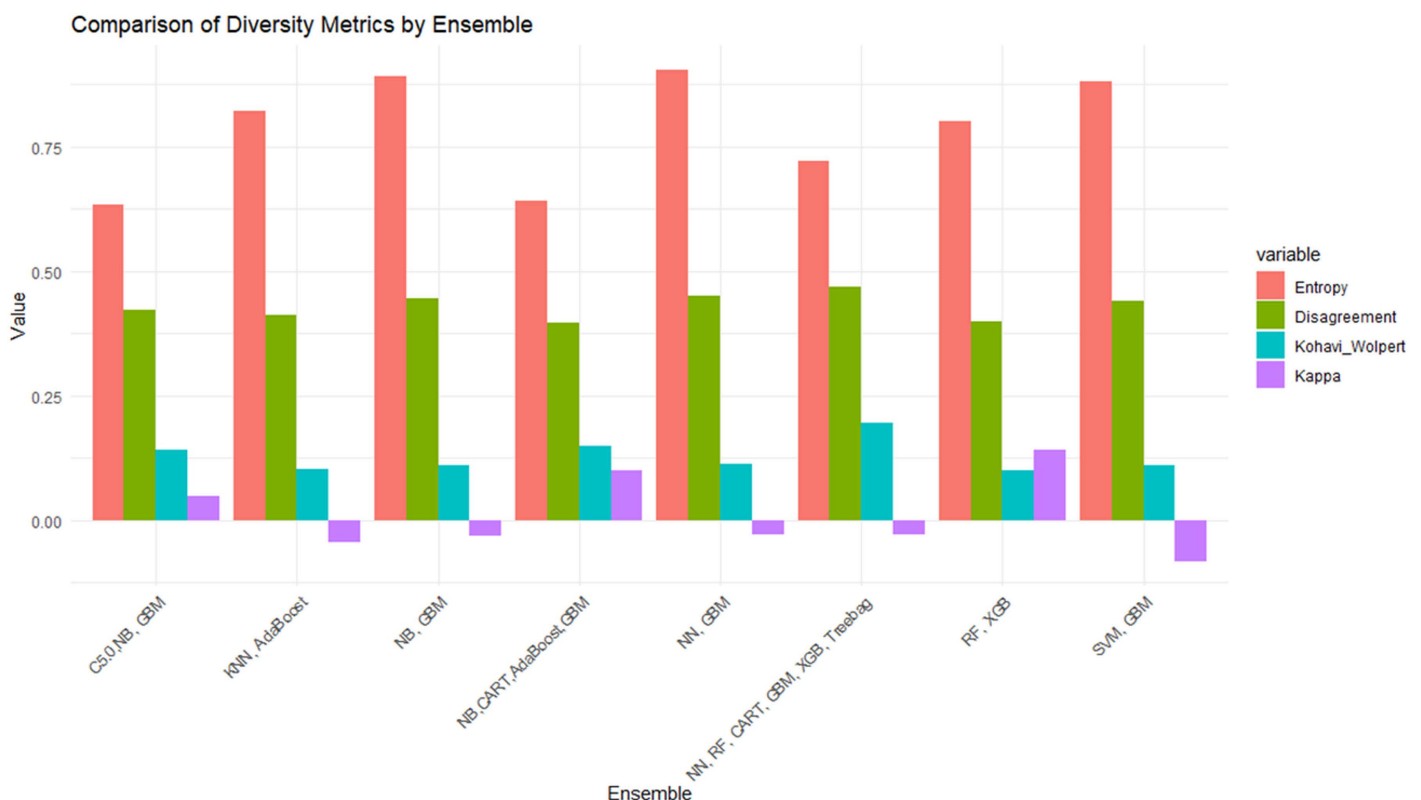

**Fig 7. Comparison of diversity metrics across eight selected sub-model sets on the test dataset.**

pairwise comparisons (e.g., GBM vs. NN–GBM stack) showed small effects ($r < 0.2$, $p > 0.05$), indicating negligible practical gain. However, combinations such as NB + GBM with GLM meta-learner, AdaBoost + KNN with GBM meta-learner, and RF + XGB with NN meta-learner achieved significant improvements ($r > 0.8$), reflecting meaningful accuracy gains.

**Table 4. Accuracy of stacking sub-model sets using five different meta-learners on the test dataset.**

| Stack Using the Linear Discriminant Analysis Meta-Learner (95% CI) | Stack using the Neural Network Meta-Learner (95% CI) | Stack Using the Gradient Boosting Machine Meta-Learner (95% CI) | Stack Using Random Forest Meta-Learner (95% CI) | Stack Using Generalized Linear Model Meta-Learner (95% CI) | Accuracy of The Best Classifier from The Ensemble (95% CI) | *Stacked Sub-Model Sets* |
|---|---|---|---|---|---|---|
| 0.826 (0.806, 0.845) | 0.823 (0.803, 0.843) | 0.826 (0.806, 0.845) | 0.809 (0.788, 0.829) | 0.803 (0.782, 0.823) | GBM: 0.825 (0.804, 0.844) | **Stacking NN, GBM** |
| 0.830 (0.809, 0.849) | 0.832 (0.812, 0.851) | 0.833 (0.813, 0.852) | 0.8144 (0.793, 0.834) | 0.836 (0.816, 0.855) | GBM: 0.825 (0.804, 0.844) | **Stacking NB, GBM** |
| 0.865 (0.846, 0.883) | 0.877 (0.859, 0.894) | 0.883 (0.865, 0.899) | 0.765 (0.742, 0.786) | 0.865 (0.846, 0.883) | SVM: 0.8332 (0.813, 0.852) | **Stacking SVM, GBM** |
| 0.848 (0.828, 0.866) | 0.840 (0.820, 0.859) | 0.876 (0.858, 0.893) | 0.852 (0.833, 0.870) | 0.848 (0.828, 0.866) | AdaBoost: 0.902 (0.885, 0.916) | **Stacking KNN, AdaBoost** |
| 0.895 (0.878, 0.910) | 0.914 (0.898, 0.928) | 0.652 (0.626, 0.676) | 0.878 (0.860, 0.894) | 0.897 (0.880, 0.913) | RF: 0.892 (0.875, 0.907) | **Stacking RF, XGB** |
| 0.898 (0.881, 0.913) | 0.904 (0.898, 0.918) | 0.903 (0.886, 0.918) | 0.909 (0.892, 0.923) | 0.895 (0.878, 0.912) | RF: 0.892 (0.875, 0.907) | **Stacking RF, CART, NN, GBM, XGB, Treebag** |
| 0.836 (0.816, 0.855) | 0.844 (0.823, 0.863) | 0.886 (0.868, 0.901) | 0.883 (0.866, 0.899) | 0.838 (0.818, 0.857) | AdaBoost: 0.902 (0.885, 0.916) | **Stacking NB,CART, GBM, AdaBoost** |
| 0.886 (0.868, 0.901) | 0.908 (0.891, 0.922) | 0.911 (0.895, 0.926) | 0.897 (0.880, 0.912) | 0.888 (0.871, 0.904) | C5.0: 0.890 (0.872, 0.905) | **Stacking NB, C5.0, GBM** |
| 0.835 (0.815, 0.854) | 0.838 (0.818, 0.857) | 0.888 (0.870, 0.903) | 0.879 (0.860, 0.895) | 0.8374 (0.817, 0.856) | AdaBoost: 0.902 (0.885, 0.916) | **Traditional stacking Ridge, KNN, CART, AdaBoost** |

The best-performing configuration—stacking RF and XGB with an NN meta-learner—achieved:

• Accuracy: 0.914 (95% CI: 0.898–0.928)

• AUC: 0.955

• F1 score: 0.801

• MCC: 0.746

This model outperformed both individual classifiers and other stacking variants (Tables 6–7, Fig 8). Wilcoxon tests showed large effect sizes ($r > 0.5$), confirming substantial performance improvements, whereas McNemar's and DeLong's tests indicated that some pairwise differences were not significant after correction. ROC curves (Fig 9) demonstrated strong sensitivity and specificity, and calibration plots (Fig 10) showed excellent alignment between predicted and observed outcomes.

### 3.6 Computational complexity and training time

Computational complexity was assessed on a standard workstation (Intel Core i5 M520 @ 2.40 GHz, 8 GB RAM, Windows 10, 64-bit). Training times varied across models: XGB required ~98 s, RF ~306 s, and AdaBoost ~869 s. The stacked RF–XGB–NN model required ~450 s—faster than AdaBoost despite its complexity (Table 8).

**Table 5. Results of significance tests comparing the best base classifier with the stacking model in each sub-model set.**

| Compared Models | Wilcoxon test | McNemar's test (df = 1) | Roc test (boot.n = 2000, boot. stratified = 1) |
|---|---|---|---|
| **For GBM vs Stacking NN, GBM using the GBM meta-learner** | V = 1982, Holm-adjusted p-value = 0.269 Effect size r = 0.107 (small) | McNemar's chi-squared = 0.304, p-value = 0.5808 | D = −1.043, p-value = 0.297 |
| **For GBM vs Stacking NB, GBM using GLM meta-learner** | V = 77, Holm-adjusted p-value = 4.36e-16 Effect size r = 0.828 (large) | McNemar's chi-squared = 3.809, p-value = 0.05096 | D = 0.708, p-value = 0.478 |
| **For SVM vs Stacking SVM, GBM using the GBM meta-learner** | V = 2087, Holm-adjusted p-value = 0.133 Effect size r = 0.151 (small) | McNemar's chi-squared = 58.21, p-value = 2.361e-14 | D = −0.816, p-value = 0.414 |
| **For AdaBoost vs Stacking KNN, AdaBoost using the GBM meta-learner** | V = 5049, Holm-adjusted p-value < 2.2e-16 Effect size r = 0.868 (large) | McNemar's chi-squared = 3.822, p-value = 0.0505 | D = −7.768, p-value = 7.952e-15 |
| **For RF vs Stacking RF, XGB using the Neural Network meta-learner** | V = 5019, Holm-adjusted p-value < 2.2e-16 Effect size r = 0.858 (large) | McNemar's chi-squared = 10.75, p-value = 0.0010 | D = −0.446, p-value = 0.6552 |
| **For RF vs Stacking RF, CART, NN, GBM, XGB, Treebag using Random Forest meta-learner** | V = 515, Holm-adjusted p-value = 4.87e-12 Effect size r = 0.691 (large) | McNemar's chi-squared = 12.41, p-value = 0.0004 | D = 0.384, p-value = 0.7007 |
| **For AdaBoost vs Stacking NB, CART, GBM, AdaBoost using the GBM meta-learner** | V = 5050, Holm-adjusted p-value < 2.2e-16 Effect size r = 0.868 (large) | McNemar's chi-squared = 3.431, p-value = 0.063 | D = −8.167, p-value = 3.161e-16 |
| **For C5.0 vs Stacking NB, C5.0, GBM using the GBM meta-learner** | V = 4922, Holm-adjusted p-value < 2.2e-16 Effect size r = 0.824 (large) | McNemar's chi-squared = 21.00, p-value = 4.581e-06 | D = −0.371, p-value = 0.71 |
| **For AdaBoost vs Traditional stacking Ridge, KNN, CART, AdaBoost using the GBM meta-learner** | V = 4533, Holm-adjusted p-value < 2.2e-16 Effect size r = 0.690 (large) | McNemar's chi-squared = 0.004, p-value = 0.9447 | D = −8.9812, p-value < 2.2e-16 |

**Table 6. Performance evaluation of selected stacking models that outperform the most accurate individual algorithm in their respective combinations.**

| Stacking Models | TN | FP | TP | FN | Accuracy (95% CI) | Kappa | Sensitivity | Specificity | Precision | F1 | MCC | ROC |
|---|---|---|---|---|---|---|---|---|---|---|---|---|
| **Stacking NB, GBM using GLM meta-learner** | 952 | 166 | 250 | 65 | 0.838 (0.818, 0.856) | 0.577 | 0.793 | 0.851 | 0.599 | 0.683 | 0.587 | 0.885 |
| **Stacking SVM, GBM using the GBM meta-learner** | 1027 | 91 | 222 | 93 | 0.872 (0.853, 0.888) | 0.625 | 0.705 | 0.919 | 0.709 | 0.707 | 0.624 | 0.893 |
| **Stacking RF, XGB using the Neural Network meta-learner** | 1063 | 55 | 247 | 68 | 0.914 (0.898, 0.928) | 0.746 | 0.784 | 0.951 | 0.818 | 0.801 | 0.746 | 0.955 |
| **Stacking RF, CART, NN, GBM, XGB, Treebag using Random Forest meta-learner** | 1061 | 57 | 241 | 74 | 0.909 (0.892, 0.923) | 0.728 | 0.765 | 0.949 | 0.809 | 0.786 | 0.728 | 0.949 |
| **Stacking NB, C5.0, GBM using the GBM meta-learner** | 1068 | 50 | 238 | 77 | 0.911 (0.895, 0.926) | 0.733 | 0.756 | 0.955 | 0.826 | 0.789 | 0.734 | 0.944 |

**Table 7. Statistical comparison between stacked random forest (RF) and XGBoost (XGB) utilizing a neural network (NN) meta-learner and other stacking models.**

| Compared Models | Wilcoxon test | McNemar's test (df = 1) | Roc test (DeLong method) |
|---|---|---|---|
| stack.GLM.GBM.NB and stack.NN.RF.XGB | V = 175, Holm-adjusted p-value < 2.2e-16 Effect size r = 0.808 (large) | McNemar's chi-squared = 58.27, p-value = 2.276e-14 | Z = −6.354, p-value = 2.099e-1 |
| stack.GBM.SVM.GBM and stack.NN.RF.XGB | V = 969, Holm-adjusted p-value < 2.2e-16 Effect size r = 0.535 (large) | McNemar's chi-squared = 0.523, p-value = 0.469 | Z = −6.036, p-value = 1.578e-0 |
| stack.GBM.KNN. AdaBoost and stack.NN.RF.XGB | V = 383, Holm-adjusted p-value = 1.80e-13 Effect size r = 0.736 (large) | McNemar's chi-squared = 2.472, p-value = 0.115 | Z = −4.485, p-value = 7.286e-0 |
| stack.RF.RF.CART. NN.GBM.XGB. Treebag and stack.NN.RF.XGB | V = 5050, Holm-adjusted p-value < 2.2e-16 Effect size r = 0.868 (large) | McNemar's chi-squared = 0.18, p-value = 0.671 | Z = −2.270, p-value = 0.0231 |
| stack.GBM. NB.C5.0.GBM and stack.NN.RF.XGB | V = 2728, Holm-adjusted p-value = 0.378 Effect size r = 0.089 (small) | McNemar's chi-squared = 2.913, p-value = 0.08783 | Z = −3.066, p-value = 0.002 |

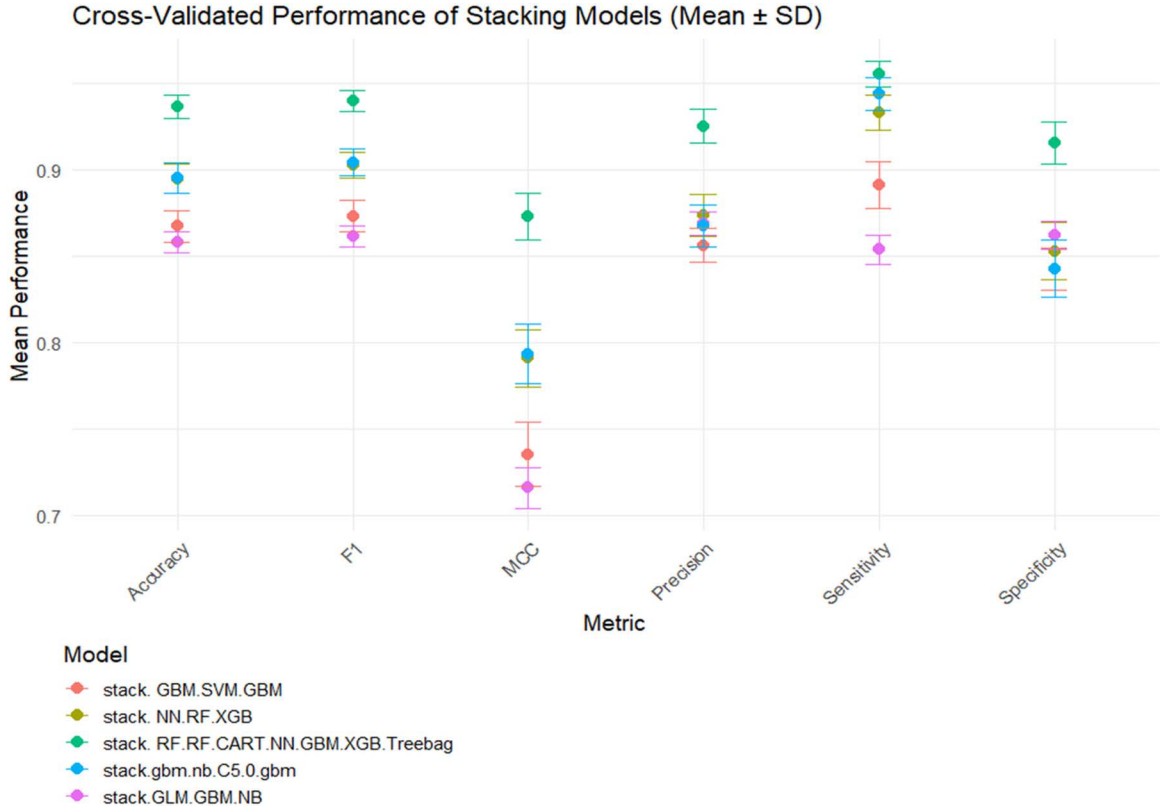

**Fig 8. Performance metrics of the selected stacked models on the training dataset.**

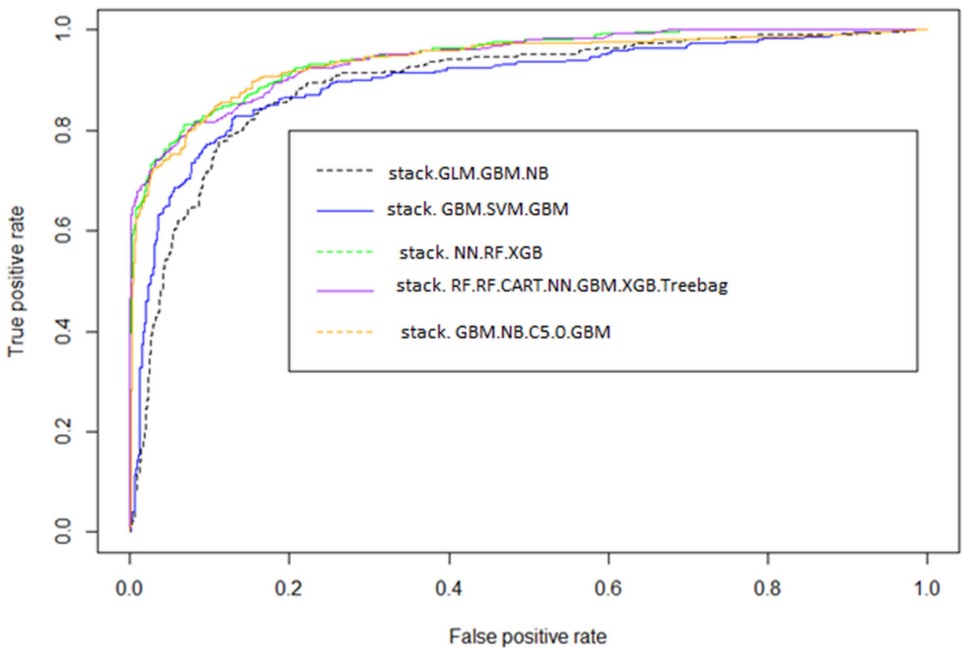

**Fig 9. ROC curves of the best-performing stacked models on the test dataset.** Stacking NB, GBM using GLM meta-learner (stack.GLM.GBM.NB). Stacking SVM, GBM using the GBM meta-learner (stack. GBM.SVM.GBM). Stacking RF, CART, NN, GBM, XGB, Treebag using Random Forest meta-learner (stack. RF.RF.CART.NN.GBM.XGB.Treebag). Stacking RF, XGB using the Neural Network meta-learner (stack. NN.RF.XGB). Stacking NB, C5.0, GBM using the GBM meta-learner (stack. GBM.NB.C5.0.GBM).

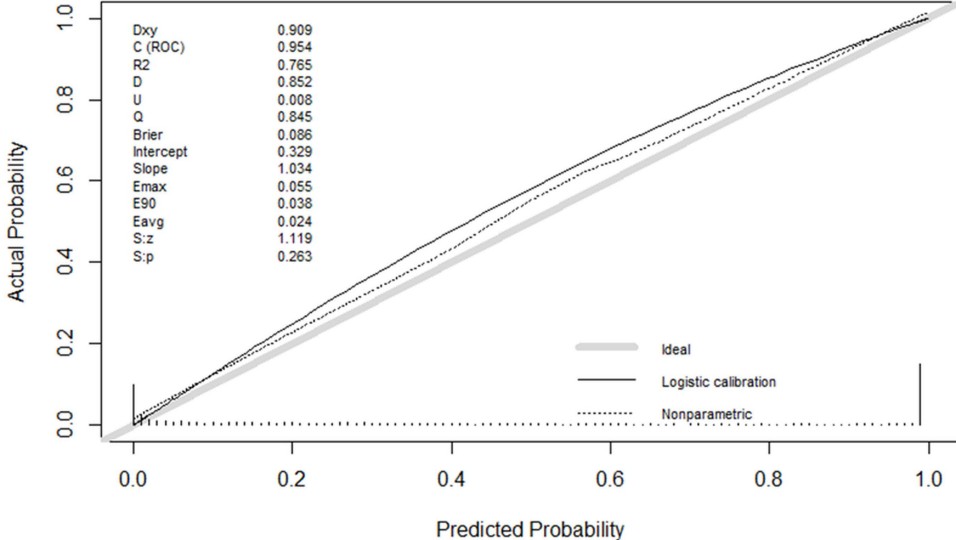

**Fig 10. Calibration plot of the stacked Random Forest (RF) and XGBoost (XGB) model using a Neural Network (NN) meta-learner under repeated 10-fold cross-validation.**

**Table 8. Training and prediction times for single models and stacking ensembles.**

| Models | Training Time (sec) | Prediction Time per Patient (sec) |
|---|---|---|
| AdaBoost | 868.6 | 0.70 |
| Random Forest (RF) | 306.5 | 0.14 |
| XGBoost (XGB) | 98.0 | 0.01 |
| Stacked RF + XGB + NN | 450.2 | 0.17 |

Inference times were uniformly low, ranging from 0.01 s per patient (XGB) to 0.70 s (AdaBoost). The stacked model achieved 0.17 s per prediction, supporting deployment in near real-time clinical settings through electronic decision-support tools.

## 3.7 Model Interpretation

Feature importance analysis of the stacked RF–XGB–NN ensemble identified age as the most influential predictor of mortality, demonstrating strong predictive stability (variability ±0.06, permutation error 0.237) (Fig 11, S5 Table in S1 File).

Neutrophil count (NEUT), phosphorus levels, and oxygen saturation while on a ventilator (O2sat.Ventilator) followed as critical predictors, reflecting infection severity, metabolic status, and respiratory function, respectively. Additional features such as lactate and ferritin contributed meaningfully, consistent with their roles in sepsis, inflammation, and critical illness.

Model-agnostic SHAP analysis, applied to the final stacked model, decomposed individual predictions into feature-level contributions. SHAP analysis indicated that for the "Death" class, advancing age ($\varphi=-0.18$), reduced O$_2$sat.Ventilator ($\varphi=-0.02$), and elevated NEUT ($\varphi=-0.08$) were dominant mortality drivers (Fig 12, S6 Table in S1 File). Reduced sodium (NA, $\varphi=-0.08$) and phosphorus (P, $\phi=-0.04$) also contributed to poor outcomes, whereas higher lactate levels had a

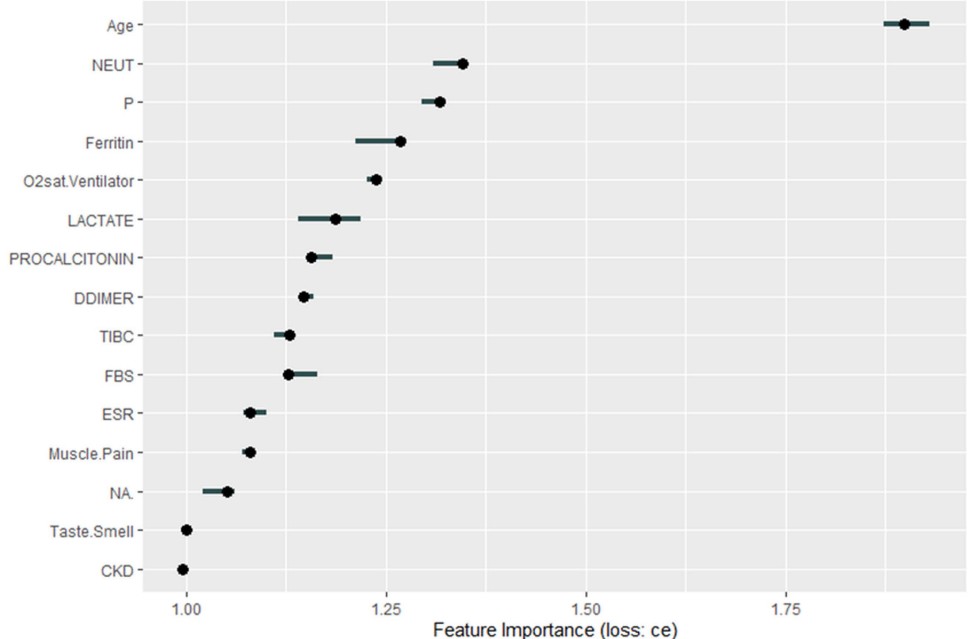

**Fig 11. Most influential predictors contributing to "Death" outcomes in the stacked RF–XGB model with an NN meta-learner.**

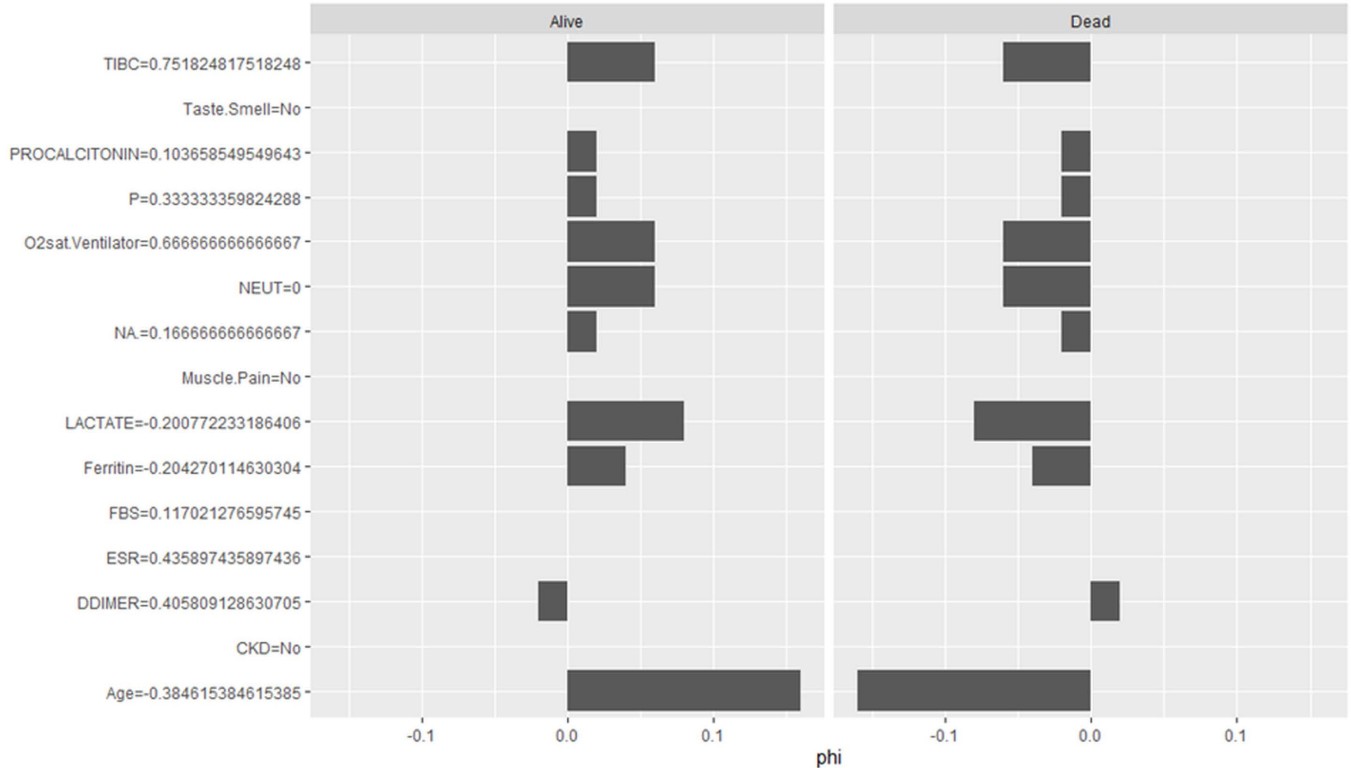

**Fig 12. SHAP-based interpretation of the stacked RF–XGB model using a Neural Network meta-learner.**

modest positive contribution ($\varphi = 0.02$). Features such as muscle pain, taste/smell disturbances, CKD, FBS, ESR, ferritin, and D-dimer had minimal SHAP contributions.

Interaction analysis (Fig 13, S7 Table in S1 File) showed that age exhibited strong interactions with $O_2$sat.Ventilator, NEUT, phosphorus, and ferritin, highlighting their synergistic effects on mortality risk. Together, these findings demonstrate that the stacking ensemble not only improves predictive accuracy but also maintains meaningful clinical interpretability.

## 4 Discussion

This study introduced a hybrid feature-selection and diversity-guided stacking framework designed to improve predictive accuracy, interpretability, and computational efficiency in high-dimensional clinical data. Although demonstrated on a large cohort of 4,778 COVID-19 patients, the proposed approach is broadly applicable to biomedical, environmental, and engineering domains that require scalable and transparent ensemble learning.

Our framework integrates hybrid feature selection—combining VIF, ANOVA, SBE, and Lasso—with a diversity-based stacking strategy that systematically quantifies inter-model complementarity using both pairwise (e.g., Yule's Q, Disagreement, Kappa) and non-pairwise (e.g., Entropy, Kohavi–Wolpert) diversity measures. This approach directly addresses major limitations of prior COVID-19 prognostic models, including small sample sizes, poor calibration, and redundant base learners [29–31]. It also mitigates the trade-offs observed in traditional models, wherein improved predictive performance often comes at the cost of computational demand or reduced interpretability—limitations frequently encountered in deep neural networks and boosting algorithms [34–37].

A key advantage of this study is the large sample size, which is substantially greater than that of many prior investigations. This enhances both the stability and generalizability of our findings. A multicenter study in Tehran reported a

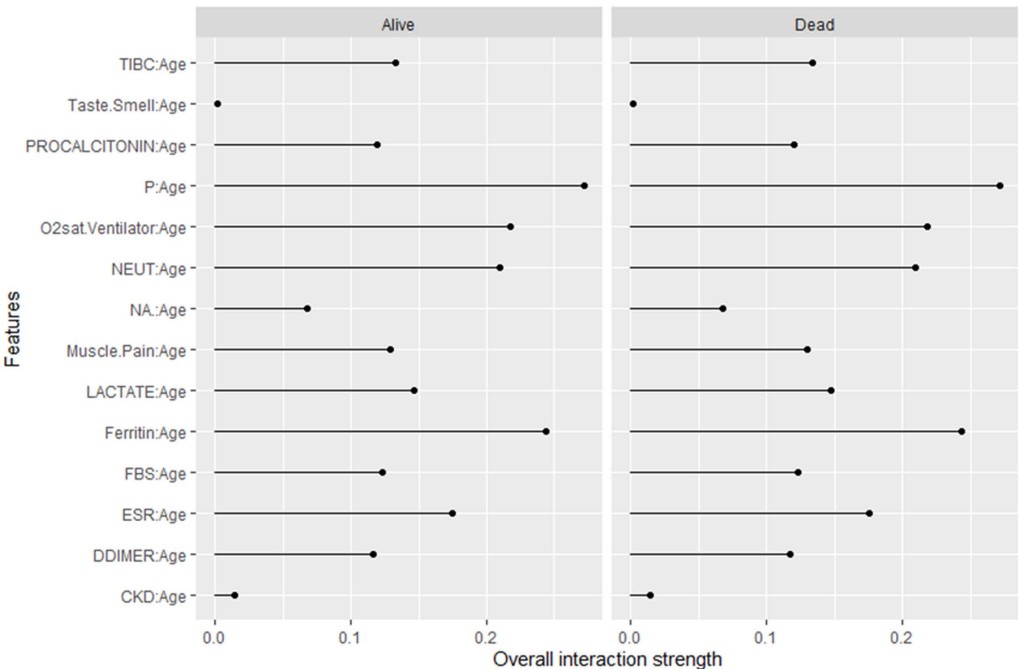

**Fig 13. Interaction effects between age and key clinical predictors influencing mortality in the stacked RF–XGB model with an NN meta-learner.**

case fatality rate (CFR) of 10.05% across 19 hospitals [73]. Consistent with guidelines recommending at least 20 events per predictor variable for outcomes with low prevalence [74], we determined that a minimum of 3,000 patients would be required to reliably identify significant mortality predictors. Our dataset exceeded this threshold, reducing the risk of over-fitting and supporting the robustness of the derived model.

The use of the Robust Scaling for standardization and ROSE-based resampling to achieve class balance were essential to model reliability where class imbalance, is a common barrier in predictive modeling [75], particularly in the context of COVID-19 [76]. The original training dataset exhibited pronounced class imbalance, with "Death" cases constituting only 22% of the cohort. This imbalance skewed model learning toward the majority "Alive" class, inflating accuracy but suppressing sensitivity—a critical limitation in mortality prediction. ROSE resampling created a fully balanced 1:1 dataset, substantially improving sensitivity and enabling models to better recognize minority-class (Death) cases. As expected, this rebalancing slightly reduced specificity due to the presence of synthetic samples. Although ROSE improves minority-class recognition and stabilizes cross-validation metrics, synthetic oversampling may introduce mild calibration shifts or artificial patterns, as noted in previous studies [77,78]. To mitigate this, final model performance was strictly evaluated on the original unbalanced test set, ensuring that reported results reflect real clinical distributions rather than synthetic balance.

Through iterative refinement, the dataset was reduced from 115 to 15 clinically interpretable features—such as age, neutrophil count, ferritin, and $O_2$ saturation—all of which are routinely available in electronic health records. These variables span immunologic, metabolic, and respiratory domains, collectively capturing key biological mechanisms underlying severe COVID-19 outcomes.

Traditional stacking methods that use highly correlated base learners provided only limited performance gains, confirming that redundancy constrains the potential benefits of ensemble modeling. In contrast, the proposed diversity-guided stacking approach achieved superior accuracy and generalization by leveraging heterogeneous classifiers with complementary error patterns.

For instance, Ribeiro et al. [79] demonstrated that stacking models can enhance predictive performance in COVID-19 outcomes. Their stack-ensemble model, which incorporated support vector regression, effectively forecasted mortality among 14,267 COVID-19 patients in Brazil. Similarly, our findings reinforce the notion that combining strong, heterogeneous classifiers through stacking is more effective than relying on a single best-performing classifier (53).

This observation aligns with emerging literature on ensemble learning. Hussain et al. [80] highlighted the superiority of hybrid classifier systems in improving prediction accuracy, and further reported an impressive AUC of 96.0% using a deep stacking neural network to predict mortality risk. Together, these studies support the effectiveness of diversity-aware ensemble approaches in high-stakes biomedical prediction.

Our stacking model significantly outperformed established machine learning approaches reported in the literature. For example, Yakovyna et al. [28] applied a combination of supervised and unsupervised learning techniques but did not achieve the level of discrimination observed in our stacking model. Rahmatinejad et al. [29] reported high Brier scores for Random Forest and improved precision and sensitivity for XGBoost; however, our RF–XGB–NN stacking configuration provided substantially higher accuracy and AUC, supported by rigorous statistical analyses, including Wilcoxon, McNemar, and DeLong tests. Furthermore, compared to a study using over 500 EHR variables to train an RF model for sepsis mortality prediction [81], our stacking method achieved notably better calibration and discrimination while maintaining computational efficiency.

Our findings also outperform those of de Paiva et al. [82], who analyzed 10,897 COVID-19 patients using various machine learning models—including FNet transformers, convolutional neural networks, support vector machines, LightGBM, and traditional statistical approaches such as LASSO and Generalized Additive Models (GAM). Their best models achieved an AUROC of 0.826 and a MacroF1 score of 65.4%. In contrast, our stacking framework delivered an F1 score of 80.1% and AUC of 0.955, demonstrating substantially improved predictive accuracy and reliability.

The Neural Network meta-learner emerged as the most effective combiner of base-model outputs. Neural meta-learning has previously been shown to outperform linear or tree-based approaches by capturing higher-order, non-linear relationships among model outputs, particularly in biomedical prediction tasks [83–85]. Our cross-validation results confirmed this, with the NN meta-learner providing superior discrimination and calibration across diverse sub-model sets.

Feature importance analysis identified age as the strongest predictor of mortality, followed by neutrophil count (NEUT), phosphorus levels, oxygen saturation ($O_2$sat.Ventilator), and lactate. Elevated NEUT counts have been widely associated with severe disease and cytokine storm responses, while reduced oxygen saturation is a direct marker of respiratory compromise [86,87]. SHAP analysis further showed that advancing age and high NEUT levels significantly increased mortality risk. These results reinforce established clinical findings that older age and immune dysregulation critically influence COVID-19 severity [88].

Beyond performance metrics, the proposed framework emphasizes computational efficiency and real-world applicability. Although training complexity increased relative to single models, the computation remained manageable and feasible for operational clinical environments. Inference latency (<0.2 seconds per prediction) is sufficiently low for real-time or near-real-time decision support, including bedside applications and automated triage systems.

Overall, this study presents a generalizable, diversity-aware ensemble framework that balances accuracy, interpretability, and computational efficiency. While validated for COVID-19 mortality prediction, the approach is adaptable to broader biomedical domains where heterogeneous data, transparency, and performance stability are critical.

## 5 Limitations and biases

This study has several limitations that should be considered when interpreting the findings. First, the retrospective EHR-based design may introduce selection and information biases, as data collection depended on available hospital records, which may not fully capture all relevant clinical variables [89]. Retrospective EHR studies are particularly vulnerable to incomplete or non-standardized data, including missing values, heterogeneous logging practices, and inconsistent variable

definitions across hospitals [90]. Moreover, implicit clinician bias, referral patterns, and disparities in diagnosis or treatment may have influenced the data used for training, thereby propagating systemic inequities into predictive algorithms [91].

Second, the dataset was limited to three Iranian hospitals, which constrains the generalizability and external validity of the model. Differences in genetic, sociodemographic, cultural, and healthcare system characteristics across populations may influence both feature distributions and outcome risks, and prior reviews have shown that COVID-19 prediction models often perform poorly outside their development setting [92]. For example, variations in comorbidity prevalence, access to intensive care resources, and laboratory reference ranges may affect model performance in non-Iranian populations. Without independent external validation, our results should be interpreted cautiously when applied elsewhere. Recent reviews further emphasize that prediction models can never be considered fully "validated," as transportability depends on population, setting, and temporal context [78,93]. Accordingly, validation in multinational, diverse cohorts is essential before clinical translation.

Third, although we applied resampling (ROSE) to mitigate class imbalance, oversampling approaches may embed artificial patterns that distort calibration or inflate predictive accuracy. This limitation has been repeatedly recognized in both COVID-19 studies and broader clinical prognostic modeling research [77,78].

Fourth, although ensemble learning (stacking) enhanced predictive performance, it introduced computational complexity and challenges for clinical deployment. Even with SHAP-based interpretability, ensemble models remain partially opaque, and post-hoc explanations represent approximations rather than causal insights. This raises concerns about clinical trust and automation bias, particularly if model outputs are adopted uncritically [93].

Fifth, our models did not incorporate unmeasured contextual confounders such as evolving treatment regimens, viral variants, or social determinants of health. As shown in previous literature, omission of such factors can bias effect estimates and limit real-world applicability [91]. Relatedly, model performance drift is a potential risk, as COVID-19 epidemiology, therapeutic strategies, and patient characteristics have evolved over time, necessitating ongoing monitoring and recalibration [78].

Finally, several design-related limitations warrant consideration. Statistical literature highlights that non-random sampling and limited site representativeness reduce the generalizability of predictive models, particularly when outcome heterogeneity is present [92]. Furthermore, most machine learning studies—including our own—focus primarily on discrimination metrics (e.g., AUC), with less emphasis on calibration and fairness assessments, which limits their clinical interpretability and adoption [93].

Despite these limitations, we employed rigorous validation strategies, including repeated cross-validation, calibration evaluation, and effect size reporting, consistent with TRIPOD recommendations [94]. These measures enhance robustness and transparency; however, independent, prospective, multi-center validation in larger and more diverse populations remains essential before clinical implementation.

## 5 Conclusion

In conclusion, this study demonstrates that a diversity-guided stacking ensemble—integrating Random Forest, XGBoost, and a Neural Network meta-learner—can achieve high predictive accuracy and interpretability for COVID-19 mortality risk. By combining a hybrid feature selection pipeline with heterogeneous base learners, the framework effectively reduced redundancy, captured nonlinear interactions, and maintained computational efficiency suitable for near real-time deployment.

Key predictors such as age, neutrophil count, phosphorus, and oxygen saturation consistently aligned with known clinical mechanisms of severe COVID-19, supporting both the statistical and biological validity of the model. SHAP-based interpretation further illustrated how interactions among these variables shape mortality risk, helping to bridge predictive performance with clinical insight.

Nevertheless, this work has several limitations, including its retrospective design, reliance on data from a single regional health system, and the absence of external validation, all of which may restrict generalizability. Future research

should extend this framework to multi-center or multi-disease cohorts, integrate multimodal data sources (e.g., imaging, genomics), and evaluate real-time performance in prospective clinical environments.

Ultimately, the proposed stacking strategy represents a scalable and interpretable modeling paradigm that can be readily adapted to a wide range of clinical prediction tasks beyond COVID-19, advancing the application of ensemble learning for precision medicine and healthcare decision support.

## Supporting information

**S1 File.** This file contains supplementary tables, figures, and additional analyses including correlation matrices, model tuning parameters, performance summaries, feature importance results, interaction analyses, and SHAP outputs. (DOCX)

## Acknowledgments

This article was part of the Ph.D. dissertation in epidemiology at Shahid Beheshti University of Medical Sciences (SBMU).

## Author contributions

**Conceptualization:** Farideh Mohtasham, Seyed Saeed Hashemi Nazari, MohamadAmin Pourhoseingholi, Kaveh Kavousi, Mohammad Reza Zali.

**Data curation:** Farideh Mohtasham, MohamadAmin Pourhoseingholi.

**Formal analysis:** Farideh Mohtasham, Seyed Saeed Hashemi Nazari, Kaveh Kavousi.

**Methodology:** Farideh Mohtasham, Seyed Saeed Hashemi Nazari, MohamadAmin Pourhoseingholi, Kaveh Kavousi, Mohammad Reza Zali.

**Project administration:** Kaveh Kavousi.

**Software:** Farideh Mohtasham, MohamadAmin Pourhoseingholi.

**Supervision:** Kaveh Kavousi, Mohammad Reza Zali.

**Validation:** Farideh Mohtasham, Seyed Saeed Hashemi Nazari, MohamadAmin Pourhoseingholi, Kaveh Kavousi, Mohammad Reza Zali.

**Visualization:** Farideh Mohtasham.

**Writing – original draft:** Farideh Mohtasham.

**Writing – review & editing:** Seyed Saeed Hashemi Nazari, MohamadAmin Pourhoseingholi, Kaveh Kavousi, Mohammad Reza Zali.

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
