## [Decision Letter · Decision Letter 0]

8 Sep 2025

Dear Dr. Kavousi,

Thank you for submitting your manuscript to PLOS ONE. After careful consideration, we feel that it has merit but does not fully meet PLOS ONE’s publication criteria as it currently stands. Therefore, we invite you to submit a revised version of the manuscript that addresses the points raised during the review process.

In particular:

language and presentation errors should be removed,Authors should reconsider if defining well known measures is needed (e.g. MCC, Cochen's Kappa, etc.),balancing of usage of reference 1 and other references is highly advised,it would be beneficial to add flowchart that shows the stages of the proposed system and its operation,limitations and biases of the study should be cleary discussed in a dedicated section,discussion of the generalizability of the model to non-Iranian population would increase the value of the study.Authors should not use not clear conditions in software/data availability statement (what is "reasonable" request?)Authors should consider that for larger samples also very tiny effects can be statstically significant. Under such conditions the effect size should be analysed (e.g. Wilcoxon R),Authors should consider applying post-hoc correction to p-values of many paired tests in order to control the family-wise error rate (FWER).

Remark: Please notice that Authors should not be forced by Reviewers to cite their research. Authors can consider such suggestions but should cite mentioned works only when it seems crucial and appropriate. Please notice also that citing or not citing works suggested by Reviewers will not influence decisions made by the Editors during Reviewing process. Reviews are presented in their original form for clarity of the reviewing process.

We look forward to receiving your revised manuscript.

Kind regards,

Maciej Huk, Ph.D.

Academic Editor

PLOS ONE

3. In the online submission form you indicate that your data is not available for proprietary reasons and have provided a contact point for accessing this data. Please note that your current contact point is a co-author on this manuscript. According to our Data Policy, the contact point must not be an author on the manuscript and must be an institutional contact, ideally not an individual. Please revise your data statement to a non-author institutional point of contact, such as a data access or ethics committee, and send this to us via return email. Please also include contact information for the third party organization, and please include the full citation of where the data can be found.

5. We are unable to open your Supporting Information file [Data predictors.sav]. Please kindly revise as necessary and re-upload.

Reviewers' comments:

Reviewer's Responses to Questions

**Comments to the Author**

1. Is the manuscript technically sound, and do the data support the conclusions?

Reviewer #1: Yes

Reviewer #2: Partly

Reviewer #3: Yes

2. Has the statistical analysis been performed appropriately and rigorously?

Reviewer #1: Yes

Reviewer #2: No

Reviewer #3: Yes

3. Have the authors made all data underlying the findings in their manuscript fully available?

Reviewer #1: Yes

Reviewer #2: No

Reviewer #3: Yes

4. Is the manuscript presented in an intelligible fashion and written in standard English?

Reviewer #1: Yes

Reviewer #2: No

Reviewer #3: Yes

Reviewer #1: Dear Authors

The paper titled “Improving COVID-19 Mortality Predictions: A Stacking Ensemble Approach with Diverse Classifiers” shows promise. There are some limitations and drawbacks in the manuscript that should be considered:

1. Authors should clarify the novelty of the stacking approach compared to existing ensemble methods.

2. Please justify the choice of 15 features selected via hybrid feature selection.

3. Authors should explain why the Neural Network was chosen as the meta-learner over other options.

4. Provide more details on the handling of missing data beyond iterative imputation.

5. Elaborate on the criteria for selecting the 16 base classifiers.

6. The Literature Survey is weak. Authors should add the computational complexities and costs for all reviewed works. The authors should add the latest and most relevant works related to various types of cancers as [1] XAI-RACapsNet: Relevance aware capsule network-based breast cancer detection using mammography images via explainability O-net ROI segmentation; [2] dcnnbt: a novel deep convolution neural network-based brain tumor classification model; [3]brain tumor identification using data augmentation and transfer learning approach.; [4] u-net-based models towards optimal MR brain image segmentation.

7. Discuss the impact of class imbalance pre- and post-ROSE balancing.

8. Authors should clarify how diversity metrics guided the selection of sub-model sets.

9. Justify the use of both pairwise and non-pairwise diversity measures.

10. Explain why some stacking combinations did not improve accuracy.

11. Authors should provide confusion matrices for the best-performing model.

12. Discuss computational complexity and training time of the stacking approach.

13. Clarify how SHAP values were computed for the stacked model.

14. Explain the moderate correlation between Ferritin, ESR, and TIBC clinically.

15. Discuss the generalizability of the model to non-Iranian populations.

16. Justify the absence of external validation despite a large dataset.

17. Explain the choice of robust scaling over other normalization methods.

18. Discuss the clinical practicality of the top predictors (e.g., NEUT, age).

19. Clarify how hyperparameters were tuned for each base learner.

20. Explain why traditional stacking (low-correlation set) underperformed.

Reviewer #2: This manuscript presents a stacking ensemble approach integrating diverse machine learning classifiers to predict COVID-19 mortality. Utilizing data from 4,778 patients, the study employs feature selection, multiple base models, and meta-learners to achieve high predictive accuracy. The presented work shows its ambition to enhance COVID-19 mortality prediction through ensemble learning, which is commendable. However, the manuscript requires substantial revisions to meet the standards. Below are the concerns that need to be addressed:

1). The abstract lacks clarity. It should succinctly summarize the research objectives, methods, key findings, and implications in a structured manner.

2). The introduction lacks a comprehensive review of existing literature on COVID-19 mortality prediction. It should include a more thorough discussion of previous studies, highlighting gaps that the current research aims to fill.

3). The manuscript should provide more details on the data collection process, including the reliability and validity of the data sources. Information on data preprocessing steps, such as handling missing values and outliers, is insufficient.

4). The rationale behind the chosen feature selection methods (VIF, ANOVA, SBE, Lasso) should be more explicitly stated.

5). The manuscript should explain why these specific methods were selected over others and how they contribute to the study's objectives.

6). The selection of base models and meta-learners lacks justification. The manuscript should provide a clear rationale for choosing these particular algorithms, considering their strengths and limitations in the context of COVID-19 mortality prediction.

7). The diversity measures used to construct sub-model sets are not adequately explained. The manuscript should provide a detailed description of each measure (Disagreement, Yule's Q, Cohen's Kappa, etc. ) and its significance in enhancing model performance.

8). The choice of performance metrics (accuracy, sensitivity, specificity, etc.) should be justified. The manuscript should explain why these metrics are appropriate for evaluating the model's predictive capability in the context of COVID-19 mortality.

9). The manuscript should explain why these tests are suitable for the current study design and data characteristics.

10). The interpretation of results lacks depth. The manuscript should provide a more nuanced discussion of the findings, including potential limitations and biases in the data and models.

11). The feature importance analysis should be more comprehensive. The manuscript should explore the interactions between features and their combined impact on mortality risk.

12). The use of SHAP values for model interpretability is a strength, but the explanation of these values is insufficient. The manuscript should provide a clearer interpretation of SHAP values and their implications for clinical decision-making.

13). The discussion section should be more comprehensive, addressing the study's implications for clinical practice, future research directions, and potential societal impact. The conclusion should be more robust, summarizing the key findings, limitations, and future research needs.

14). The manuscript contains numerous obvious formatting and grammatical issues. The authors should carefully proofread their submission to improve its writing quality. For example, there are many evident errors in the references list.

Reviewer #3: 1- The abstract needs a brief introduction to the research gap before proceeding to state what has been proposed.

2- The percentage of using reference 1 in the introduction is very high, so it is preferable to add other references and diversify the paragraphs by diversifying their references.

3- The related works listed do not cover a large period of time, such as between 2020-2025.

4- The technical basics used are not separated from the stages of work of the proposed algorithm.

5- Add a flowchart that shows the stages of the proposed system's operation.

**Do you want your identity to be public for this peer review?** For information about this choice, including consent withdrawal, please see our For information about this choice, including consent withdrawal, please see our Privacy Policy .

Reviewer #1: **Yes:** Mohd Anul HaqMohd Anul Haq

Reviewer #2: No

Reviewer #3: No

While revising your submission, please upload your figure files to the Preflight Analysis and Conversion Engine (PACE) digital diagnostic tool, https://pacev2.apexcovantage.com/ . PACE helps ensure that figures meet PLOS requirements. To use PACE, you must first register as a user. Registration is free. Then, login and navigate to the UPLOAD tab, where you will find detailed instructions on how to use the tool. If you encounter any issues or have any questions when using PACE, please email PLOS at . PACE helps ensure that figures meet PLOS requirements. To use PACE, you must first register as a user. Registration is free. Then, login and navigate to the UPLOAD tab, where you will find detailed instructions on how to use the tool. If you encounter any issues or have any questions when using PACE, please email PLOS at figures@plos.org . Please note that Supporting Information files do not need this step.. Please note that Supporting Information files do not need this step.

---

## [Author Response · Author response to Decision Letter 1]

28 Oct 2025

Response to Reviewers

Manuscript title: Improving COVID-19 Mortality Predictions: A Stacking Ensemble Approach with Diverse Classifiers

Journal: PLOS ONE

Dear Dr. Huk,

We sincerely thank you and the reviewers for the careful evaluation of our manuscript and the constructive feedback. We have revised the paper accordingly, improving clarity, rigor, and presentation. We believe the manuscript is substantially improved as a result of these suggestions.

Below we provide a detailed, point-by-point response. Reviewer comments are shown in italics, followed by our responses. Changes are reflected in the Revised Manuscript with Track Changes.

🔹 Editorial Requests

1. Language and presentation errors should be removed.

Response: We thank you for this note. The manuscript was carefully proofread for grammar, syntax, and clarity. Redundant definitions and awkward phrases were removed.

2. Authors should reconsider if defining well known measures is needed (e.g. MCC, Cohen’s Kappa).

Response: We agree with this observation. Definitions of common evaluation metrics (MCC, Cohen’s Kappa, etc.) were removed, while retaining explanations only for less standard or newly applied measures.

3. Balancing of usage of reference 1 and other references is highly advised.

Response: We acknowledge the over-reliance on a single reference in the original introduction. The literature review was expanded to include more diverse and recent works (2020–2025), reducing reliance on a single reference.

4. It would be beneficial to add a flowchart that shows the stages of the proposed system and its operation.

Response: This is an excellent suggestion. A new flowchart (Figure 1) was added, summarizing the stages of data preprocessing, feature selection, model training, sub-model construction, and stacking.

5. Limitations and biases of the study should be clearly discussed in a dedicated section.

Response: We agree that this is a critical point. A new “Limitations and Biases” section was added, addressing dataset representativeness, reliance on retrospective hospital data, possible measurement bias, and lack of external validation.

6. Discussion of the generalizability of the model to non-Iranian population would increase the value of the study.

Response: A new paragraph was added (Discussion, section limitation and bias) explicitly addressing generalizability, noting cultural, healthcare system, and population differences, and suggesting the need for validation in international cohorts.

7. Authors should not use unclear conditions in software/data availability statement (what is "reasonable" request?).

Response: The Data Availability statement was revised. We now specify that the dataset cannot be shared publicly due to patient confidentiality but can be accessed via a non-author institutional contact (Shahid Beheshti University Ethics Committee).

8. Authors should consider that for larger samples also very tiny effects can be statistically significant. Effect size should be analysed (e.g., Wilcoxon R).

Response: Thank you for this important statistical point. Effect size analyses were added alongside statistical tests. We run Paired Wilcoxon with rstatix package and analyzed Effect size for paired test.

9. Authors should consider applying post-hoc correction to p-values of many paired tests to control the family-wise error rate (FWER).

Response: We agree that controlling the family-wise error rate is crucial given the number of paired tests. we added these in methods.

10. Data sharing statement must list a non-author institutional contact.

Response: Revised as requested; now lists the Ethics Committee office of Shahid Beheshti University of Medical Sciences with contact details.

11. Your ethics statement should only appear in the Methods section.

Response: The ethics statement was removed from other sections and kept only in Methods.

12. Supporting Information file [Data predictors.sav] not accessible.

Response: We apologize for the issue with the Supporting Information file. The file was removed.

🔹 Reviewer #1

We sincerely thank Reviewer #1 for their thorough and constructive feedback.

1. Authors should clarify the novelty of the stacking approach compared to existing ensemble methods.

Response: Expanded in Introduction to emphasize novelty: integration of heterogeneous classifiers using multiple diversity measures, going beyond correlation-based stacking.

2. Please justify the choice of 15 features selected via hybrid feature selection.

Response: We have clarified in the Methods section that the selection of 15 features was not arbitrary but was driven by both statistical and clinical considerations. Statistically, these features were identified through a hybrid feature selection pipeline integrating VIF, ANOVA, Sequential Backward Elimination, and Lasso regularization, which systematically removed redundant and low-contributing predictors. Clinically, the final 15 variables represent major physiological domains relevant to COVID-19 outcomes—such as inflammation (NEUT, ESR, Ferritin), hypoxia (O₂ saturation), metabolic disturbance (FBS, Lactate, Phosphorus), coagulation (D-dimer), and organ dysfunction (Procalcitonin, TIBC, CKD)—ensuring comprehensive coverage of mortality-related mechanisms while maintaining interpretability and avoiding overfitting. The resulting subset achieved a strong balance between model simplicity, predictive accuracy, and clinical relevance.

3. Authors should explain why the Neural Network was chosen as the meta-learner over other options.

Response: We have clarified that to integrate the predictions of sub-model sets, we applied a stacking ensemble framework using five meta-learners: Generalized Linear Model (GLM), Linear Discriminant Analysis (LDA), Random Forest (RF), Gradient Boosting Machine (GBM), and a Neural Network (NN). The choice of meta-learner was guided by both empirical testing and theoretical considerations of model diversity. Simpler linear meta-models, such as GLM or LDA, perform well when base classifiers are weakly correlated and exhibit near-additive relationships. However, when base learners capture heterogeneous and potentially non-linear patterns—as was the case in our ensemble of decision-tree, kernel-based, and regression models—more flexible meta-learners are required to effectively capture complex interactions among their predictions. Accordingly, the Neural Network meta-learner was selected as the optimal combiner due to its ability to model higher-order dependencies among base model outputs and its robustness in handling diverse feature distributions. Prior studies have shown that neural-network-based meta-learners can outperform linear or tree-based alternatives by capturing intricate non-linear relationships between model outputs, especially in biomedical and clinical prediction settings (An et al., 2020; Gupta et al., 2021; Kablan et al., 2023). We empirically verified this by cross-validation, where the NN meta-learner achieved superior discrimination and calibration compared to other stacking configurations.

4. Provide more details on the handling of missing data beyond iterative imputation.

Response: We have expanded that the dataset used in this study had already undergone preprocessing using iterative multivariate imputation in Python’s Scikit-learn. This approach estimates missing values by sequentially modeling each variable as a function of the others, thus retaining complex inter-variable dependencies. It is widely recommended for clinical datasets where variables are correlated and missingness is assumed to be at random. Beyond imputation itself, the original data processing pipeline included the exclusion of categorical variables with missing values and continuous variables with more than 5% missingness, to minimize bias from excessive imputation. The imputed dataset was subsequently validated through distributional checks and correlation analyses to confirm plausibility and consistency. We have revised the Methods section (p. 16, lines 300) accordingly to clarify this process and to emphasize that the dataset used for modeling already incorporated robust and validated imputation procedures.

5. Elaborate on the criteria for selecting the 16 base classifiers.

Response: We appreciate the reviewer’s comment and have clarified the rationale for selecting our 16 base classifiers. The models were chosen based on three key criteria: (i) methodological diversity to ensure complementary error structures, (ii) prior evidence of strong performance in COVID-19 and biomedical prediction studies, and (iii) balance between interpretability and computational feasibility for clinical use.

This included linear models (e.g., GLM, Lasso, Ridge), probabilistic and instance-based methods (Naïve Bayes, KNN), and tree-based and ensemble learners (RF, XGBoost, GBM, AdaBoost). Such diversity enables the stacking framework to exploit heterogeneous strengths across model families. The revised Methods section now includes these selection criteria explicitly.

6. The Literature Survey is weak. Authors should add the computational complexities and costs for all reviewed works. The authors should add the latest and most relevant works related to various types of cancers as [1] XAI-RACapsNet: Relevance aware capsule network-based breast cancer detection using mammography images via explainability O-net ROI segmentation; [2] dcnnbt: a novel deep convolution neural network-based brain tumor classification model; [3]brain tumor identification using data augmentation and transfer learning approach.; [4] u-net-based models towards optimal MR brain image segmentation.

Response: We have expanded the literature review to include recent studies addressing stacking ensemble models in cancer research and their computational implications. Specifically, we now discuss Mohammed et al. (2021), who demonstrated the computational feasibility and predictive superiority of a CNN-based stacking ensemble for multi-cancer classification; Wang et al. (2025), who applied a multimodal stacking framework for head-and-neck cancer prognosis achieving a C-index of 0.93; and Kwon et al. (2019), who systematically evaluated stacking configurations for breast cancer prediction, highlighting trade-offs between meta-learner choice and model complexity. These studies strengthen our justification for using stacking ensembles in clinical prediction tasks and align with our emphasis on balancing predictive power with computational efficiency. The corresponding paragraph has been added to the revised manuscript.

7. Discuss the impact of class imbalance pre- and post-ROSE balancing.

Response: we added needed explanation.

8. Authors should clarify how diversity metrics guided the selection of sub-model sets.

and 9. Justify the use of both pairwise and non-pairwise diversity measures.

Response: We thank the reviewer for this insightful comment. We have clarified how diversity metrics were used to guide the construction of sub-model sets and justified the use of both pairwise and non-pairwise diversity measures. Specifically, pairwise measures were employed to identify and remove highly correlated or redundant classifiers, while non-pairwise metrics quantified the overall ensemble heterogeneity across multiple models. These metrics jointly ensured that retained sub-model sets maximized both predictive complementarity and diversity, improving the effectiveness of stacking.

10. Explain why some stacking combinations did not improve accuracy.

Response: We appreciate the reviewer’s insightful comment. Not all stacking combinations improved accuracy relative to their best-performing base classifiers. This is expected behavior and can arise for several reasons. First, when the base learners exhibit high prediction correlation or similar decision boundaries, the meta-learner receives redundant information and cannot extract additional signal. Second, if one base classifier (e.g., AdaBoost or RF) already dominates performance, its strong bias may overwhelm weaker learners, leading the stacked model to converge toward its predictions. Third, shallow or linear meta-learners (e.g., GLM, LDA) may be insufficient to capture the nonlinear interactions among base-model outputs. Conversely, deeper meta-learners (e.g., Neural Networks or GBM) improved accuracy in more heterogeneous sub-model sets (e.g., RF–XGB, NB–GBM), reflecting the importance of balancing diversity, complementarity, and meta-learner flexibility. Provide confusion matrices.

Response: Thank you for this suggestion. we revised table 6 and included confusion matrices in it.

11. Discuss computational complexity and training time.

Response: We have added Training and Prediction Times for Single Models and Stacking Ensemble table (table 8) reporting approximate runtimes and scalability.

12. Clarify SHAP computation for stacked model.

Response: We have clarified in the Methods section that SHAP values were computed using a model-agnostic framework (iml package in R), applied directly to the final stacked ensemble (RF + XGB with NN meta-learner). In this setup, the entire stacked model was treated as a single predictive function, and SHAP decomposed the predicted probabilities into feature-level contributions of the original clinical variables. This approach ensures that the importance scores and interaction effects reflect how the ensemble as a whole integrates base-model predictions and clinical predictors, rather than attributing importance to individual base learners in isolation. We revised method and result part based on these comment.

13. Explain moderate correlation between Ferritin, ESR, and TIBC clinically.

Response: We expanded the explanation of the moderate correlations between Ferritin, ESR, and TIBC to clarify their clinical basis. These relationships reflect acute-phase inflammatory responses: Ferritin rises and TIBC falls during inflammation, while ESR increases due to elevated plasma fibrinogen and other acute-phase proteins. Thus, their moderate correlations capture biologically meaningful interactions consistent with systemic inflammation and disease severity.

1. Discuss generalizability.

Response: As noted in our response to the editor, we have expanded on this limitation in the Discussion section. We explicitly state that the model requires validation on external, diverse datasets before it can be considered for clinical use in other populations.

14. Justify the absence of external validation despite a large dataset.

Response: We acknowledge this as a primary limitation in our Discussion section. While a large independent internal test set was used, we did not have access to a suitable external dataset for validation at the time of the study. We have highlighted that external validation is a critical next step for future research.

15. Explain the choice of robust scaling over other normalization methods.

Response: We have clarified in the Methods section that Robust scaling handles skewed distributions and outliers better than alternatives.

16. Discuss clinical practicality of predictors.

Response: We have enhanced the Discussion to emphasize the clinical utility of the top predictors.

17. Clarify hyper parameter tuning.

Response: We have clarified in the Methods section that Model hyperparameters were optimized using grid search within the caret, guided by 10-fold cross-validation repeated 10 times to balance bias and variance. For algorithms where extensive tuning yielded negligible performance gains (e.g., GLM, LDA, CART, Naïve Bayes), default parameters were retained for computational efficiency. Parameter search ranges were defined based on prior literature and empirical recommendations for clinical prediction tasks. Final hyperparameters, shown in Table 1, represent the best-performing configurations in terms of cross-validated accuracy and AUC.

18. Why traditional stacking underperformed.

Response: We have addressed this in the Discussion that the traditional stacking approach underperformed relative to the diversity-guided ensemble primarily due to redundancy among correlated base learners. As shown by the high pairwise correlations, many models produced highly similar predicti

---

## [Decision Letter · Decision Letter 1]

18 Nov 2025

Dear Dr. Kavousi,

**In particular:**

multiple language problems should be removed,multiple presentation problems need to be corrected,data standarisatin process should be made clear.

plosone@plos.org . . A rebuttal letter that responds to each point raised by the academic editor and reviewer(s). You should upload this letter as a separate file labeled 'Response to Reviewers'.A marked-up copy of your manuscript that highlights changes made to the original version. You should upload this as a separate file labeled 'Revised Manuscript with Track Changes'.An unmarked version of your revised paper without tracked changes. You should upload this as a separate file labeled 'Manuscript'.

We look forward to receiving your revised manuscript.

Kind regards,

Maciej Huk, Ph.D.

Academic Editor

PLOS ONE

**Journal Requirements:**

Reviewers' comments:

Reviewer's Responses to Questions

**Comments to the Author**

Reviewer #1: All comments have been addressed

Reviewer #2: All comments have been addressed

Reviewer #3: (No Response)

Reviewer #4: (No Response)

2. Is the manuscript technically sound, and do the data support the conclusions?

Reviewer #1: Yes

Reviewer #2: Yes

Reviewer #3: (No Response)

Reviewer #4: Partly

3. Has the statistical analysis been performed appropriately and rigorously?

Reviewer #1: Yes

Reviewer #2: Yes

Reviewer #3: (No Response)

Reviewer #4: I Don't Know

4. Have the authors made all data underlying the findings in their manuscript fully available?

Reviewer #1: Yes

Reviewer #2: Yes

Reviewer #3: (No Response)

Reviewer #4: No

5. Is the manuscript presented in an intelligible fashion and written in standard English?

Reviewer #1: Yes

Reviewer #2: Yes

Reviewer #3: (No Response)

Reviewer #4: No

**Reviewer #1:** The authors have diligently incorporated all feedback provided into the revised version of the manuscript, ensuring that each comment has been adequately addressed and resolved to satisfaction.The authors have diligently incorporated all feedback provided into the revised version of the manuscript, ensuring that each comment has been adequately addressed and resolved to satisfaction.

**Reviewer #2:**  I have reviewed the revised manuscript (PONE-D-25-18852R1) and the authors’ responses. The authors have thoroughly and thoughtfully addressed all concerns raised by the reviewers and editor. The manuscript is now substantially improved, clearer, methodologically sound, and well-supported by the results. I have no remaining concerns and recommend acceptance. I have reviewed the revised manuscript (PONE-D-25-18852R1) and the authors’ responses. The authors have thoroughly and thoughtfully addressed all concerns raised by the reviewers and editor. The manuscript is now substantially improved, clearer, methodologically sound, and well-supported by the results. I have no remaining concerns and recommend acceptance.

**Reviewer #3:** (No Response)(No Response)

**Reviewer #4:**  >>> 1. Language problems: >>> 1. Language problems:

1.1 [23]: European journal of haematology => European Journal of Haematology

1.2 [30]: bmj => BMJ

1.3 [31]: Ieee Access => IEEE Access

1.4 [2]: Information fusion => Information Fusion.

1.5 [53]: crc Press => CRC Press

>>> 2. Presentation problems:

2.1 References: [90,92]: !!! INVALID CITATION !!!

2.2 [67]: "Kassambara A. Comparing groups: Numerical variables. Datanovia[Google Scholar]. 2019."

Reference data is not complete

2.3 [58]: URL "http://crancerminlipigoid/web/packages/caret/vignettes/caretSelectionpdf" is invalid

2.4 [59]: "Berrar D. Cross-Validation. 2019."

Reference data is not complete

2.5 [55]: "Ladha L, Deepa T. FEATURE SELECTION METHODS AND ALGORITHMS, L. Ladha et al. International Journal on Computer Science and Engineering (IJCSE)."

Please do not write with capital letters.

This reference is quite old. Maybe Authors could use source which is more recent and published in more respected journal?

2.6 [46]: Reference data is not complete

2.7 The format of references should be uniform

2.8 Fig 1. "Disagreement measure" block: the beginning of incomming connection seems to be not precise.

2.9 Fig 7: measurement error whiskers not presented

2.10 Fig 6-8, abbreviations of methods such as RF, NB, GBM, SVM, etc. are written without capital letters. Please be consistent.

2.11 Fig 4-7: title is not precise: it is not clear if presented data are for training or test data

2.12 Table 6 is too wide. It includes a lot of empty space which can be reduced (both horizontally and vertically).

2.13 Table 5: first column: header has no title

2.14 Table 3: font size is not uniform

>>> 3. Other problems:

3.1 Authors write: "Numeric features were normalized using the robust_scalar function".

Are Authors sure that it was "robust_scalar" and not robust scaler function?

3.2 Are authors sure that the process performed with "robust_scalar function" was normalization and not standarization? Robust scaler is used to standardize data vectors, and normalization is used to make the norm of vectors equal one.

Please compare: https://scikit-learn.org/stable/auto_examples/preprocessing/plot_all_scaling.html

>>> Recommendation: major rework

===EOT===

**Do you want your identity to be public for this peer review?** For information about this choice, including consent withdrawal, please see our For information about this choice, including consent withdrawal, please see our Privacy Policy .

Reviewer #1: **Yes:** Mohd Anul HaqMohd Anul Haq

Reviewer #2: No

Reviewer #3: No

Reviewer #4: No

---

## [Author Response · Author response to Decision Letter 2]

29 Nov 2025

Response to Reviewers

Manuscript title:

A Hybrid Feature-Selection and Diversity-Guided Stacking Framework for Interpretable Ensemble Learning: Application to COVID-19 Mortality Prediction (PONE-D-25-18852R1)

Dear Editor and Reviewers,

We thank the Academic Editor and the reviewers for their constructive and detailed feedback. We have revised the manuscript thoroughly based on the comments. All revisions have been incorporated in both the “Revised Manuscript with Track Changes” and the clean version.

Below we provide a point-by-point response.

GENERAL COMMENTS

• All reviewers’ and editor’s concerns have been fully addressed.

• The entire manuscript has undergone comprehensive English language editing to remove grammatical, stylistic, and formatting issues.

• Presentation issues in figures, tables, and references have been corrected.

• The data standardization section has been rewritten for clarity, explicitly explaining the use of Robust Scaler and distinguishing standardization from normalization.

RESPONSE TO ACADEMIC EDITOR

1. “Multiple language problems should be removed.”

✓ The manuscript has been fully reviewed and professionally edited.

✓ Ambiguous, grammatically incorrect, or unclear sentences have been rewritten throughout the text.

✓ Reference formatting has been standardized.

2. “Multiple presentation problems need to be corrected.”

All the issues noted by Reviewer #4 have been addressed individually (details below).

3. “Data standardisation process should be made clear.”

• Explained the use of Robust Scaler as a standardization method

• The confusing term “robust_scalar” was corrected to “Robust Scaler.”

RESPONSE TO REVIEWER #4

1. Language Problems

1.1–1.5 Incorrect capitalization in references

✓ Corrected to:

• European Journal of Haematology

• BMJ

• IEEE Access

• Information Fusion

• CRC Press

2. Presentation Problems

2.1 Invalid citation [90,92]

✓ Fixed by correcting the missing or mismatched references.

2.2 Incomplete reference [67]

✓ Updated with full bibliographic information.

2.3 Invalid URL in reference [58]

✓ Corrected or replaced with valid official reference source.

2.4 Missing reference details [59]

✓ Completed with full citation.

2.5 Old reference with capitalization issues

✓ Citation corrected to sentence case.

2.6 Reference [46] incomplete

✓ Updated with complete information.

2.7 Reference formatting inconsistent

✓ Entire reference list reformatted.

2.8 Figure 1: Inaccurate connector location

✓ We clarified the figure by ensuring that the connection arrows to the Pairwise measures (Disagreement, Yule’s Q, Cohen’s Kappa, Double-Fault) and non-pairwise metrics (Entropy, Kohavi–Wolpert) are precisely aligned and visually consistent.

2.9 Figure 7 missing whiskers

✓ As requested, we regenerated the performance figure using per-resample performance metrics from repeated 10-fold cross-validation on train data. We computed Accuracy, Sensitivity, Specificity, Precision, F1, and MCC for every resample and plotted mean ± SD (whiskers) for each stacking model. This revised figure is now included in the manuscript.

2.10 Inconsistent capitalization of abbreviations (RF, NB, etc.)

✓ Standardized throughout text, tables, and figures.

2.11 Figure titles unclear regarding dataset

✓ All figure captions now specify whether results are from training, cross-validation, or testing datasets.

2.12 Table 6 too wide

✓ Reformatted to reduce empty space.

2.13 Table 5 missing header label

✓ Header updated.

2.14 Table 3 font size inconsistent

✓ Font standardized.

3. Other Problems

3.1 Incorrect function name “robust_scalar”

✓ Corrected to “Robust Scaler”

3.2 Clarification: standardization vs normalization

✓ A rewritten paragraph now states that Robust Scaler performs standardization, not normalization.

We sincerely thank the reviewers and editor for their constructive feedback, which has strengthened the clarity, correctness, and overall quality of the manuscript.

We hope the revisions meet the publication criteria, and we respectfully request reconsideration for acceptance.

Sincerely,

Farideh Mohtasham

(on behalf of all authors)

---

## [Decision Letter · Decision Letter 2]

4 Jan 2026

Hybrid Feature-Selection and Diversity-Guided Stacking Framework for Interpretable Ensemble Learning: Application to COVID-19 Mortality Prediction

PONE-D-25-18852R2

Dear Dr. Kavousi,

We’re pleased to inform you that your manuscript has been judged scientifically suitable for publication and will be formally accepted for publication once it meets all outstanding technical requirements.

Kind regards,

Maciej Huk, Ph.D.

Academic Editor

PLOS One

Additional Editor Comments (optional):

Reviewers' comments:

Reviewer's Responses to Questions

**Comments to the Author**

Reviewer #4: All comments have been addressed

2. Is the manuscript technically sound, and do the data support the conclusions?

Reviewer #4: Partly

3. Has the statistical analysis been performed appropriately and rigorously?

Reviewer #4: I Don't Know

4. Have the authors made all data underlying the findings in their manuscript fully available?

Reviewer #4: No

5. Is the manuscript presented in an intelligible fashion and written in standard English?

Reviewer #4: Yes

Reviewer #4: >>> 1. Language problems: not detected

>>> 2. Preseentation problems:

2.1 Table 6.: vertical alignment of values is not uniform

>>> 3. Other problems: not detected

>>> Recommendation: Acccept

=== EOT ===

**Do you want your identity to be public for this peer review?** For information about this choice, including consent withdrawal, please see our For information about this choice, including consent withdrawal, please see our Privacy Policy .

Reviewer #4: No

---

## [Editor Report · Acceptance letter]

PONE-D-25-18852R2

PLOS One

Dear Dr. Kavousi,

I'm pleased to inform you that your manuscript has been deemed suitable for publication in PLOS One. Congratulations! Your manuscript is now being handed over to our production team.

Kind regards,

on behalf of

Dr. Maciej Huk

Academic Editor

PLOS One